# Spatial and Bioaccumulation of Heavy Metals in a Sheep-Based Food System: Implications for Human Health

**DOI:** 10.3390/toxics12100752

**Published:** 2024-10-16

**Authors:** Florin-Ioan Fechete, Maria Popescu, Sorin-Marian Mârza, Loredana-Elena Olar, Ionel Papuc, Florin-Ioan Beteg, Robert-Cristian Purdoiu, Andrei Răzvan Codea, Caroline-Maria Lăcătuș, Ileana-Rodica Matei, Radu Lăcătuș, Adela Hoble, Ioan Valentin Petrescu-Mag, Florin-Dumitru Bora

**Affiliations:** 1Clinical Sciences Department, Faculty of Veterinary Medicine, University of Agricultural Sciences and Veterinary Medicine Cluj-Napoca, 3-5 Mănăștur Street, 400372 Cluj-Napoca, Romania; florin-ioan.fechete@student.usamvcluj.ro (F.-I.F.); sorin.marza@usamvcluj.ro (S.-M.M.); loredana.olar@usamvcluj.ro (L.-E.O.); robert.purdoiu@usamvcluj.ro (R.-C.P.); razvan.codea@usamvcluj.ro (A.R.C.); caroline-maria.lacatus@student.usamvcluj.ro (C.-M.L.); radu.lacatus@usamvcluj.ro (R.L.); 2Equine Clinic, Faculty of Veterinary Medicine, University of Agricultural Sciences and Veterinary Medicine Cluj-Napoca, 3-5 Mănăștur Street, 400372 Cluj-Napoca, Romania; maria.popescu@usamvcluj.ro; 3Preclinic Department, Faculty of Veterinary Medicine, University of Agricultural Sciences and Veterinary Medicine Cluj-Napoca, 3-5 Mănăştur Street, 400372 Cluj-Napoca, Romania; ionel.papuc@usamvcluj.ro; 4Clinical and Paraclinical Sciences Department, Faculty of Veterinary Medicine, University of Agricultural Sciences and Veterinary Medicine Cluj-Napoca, 3-5 Mănăştur Street, 400372 Cluj-Napoca, Romania; florin.beteg@usamvcluj.ro; 5Plastic Surgery Department, University of Medicine and Pharmacy, 8 Victor Babes Street, 400012 Cluj-Napoca, Romania; irmatei@yahoo.com; 6Plastic Surgery Clinic, Spitalul Clinic de Recuperare, 46-50 Viilor Street, 400347 Cluj-Napoca, Romania; 7Research Laboratory Regarding Exploitation of Land Improvement, Land Reclamation Systems and Irrigation of Horticultural Crops, Advanced Horticultural Research Institute of Transylvania, Faculty of Horticulture and Business in Rural Development, University of Agricultural Sciences and Veterinary Medicine Cluj-Napoca, 3-5 Mănăștur Street, 400372 Cluj-Napoca, Romania; adela.hoble@usamvcluj.ro; 8Department of Environmental Engineering and Protection, Faculty of Agriculture, University of Agricultural Sciences and Veterinary Medicine Cluj-Napoca, 3-5 Mănăștur Street, 400372 Cluj-Napoca, Romania; ioan.mag@usamvcluj.ro; 9Bioflux SRL, 54 Ceahlău Street, Cluj-Napoca, 400488 Cluj-Napoca, Romania; 10Doctoral School of Engineering, University of Oradea, 1 Universității Street, 410087 Oradea, Romania; 11Viticulture and Oenology Department, Advanced Horticultural Research Institute of Transylvania, Faculty of Horticulture and Business in Rural Development, University of Agricultural Sciences and Veterinary Medicine Cluj-Napoca, 3-5 Mănăștur Street, 400372 Cluj-Napoca, Romania; 12Laboratory of Chromatography, Advanced Horticultural Research Institute of Transylvania, Faculty of Horticulture and Business for Rural Development, University of Agricultural Sciences and Veterinary Medicine, 400372 Cluj-Napoca, Romania

**Keywords:** food safety, environmental monitoring, trace elements, sheep husbandry, risk assessment

## Abstract

Heavy metal contamination in agricultural soils presents serious environmental and health risks. This study assessed the bioaccumulation and spatial distribution of nickel, cadmium, zinc, lead, and copper within a sheep-based food chain in the Baia Mare region, Romania, which includes soil, green grass, sheep serum, and dairy products. Using inductively coupled plasma mass spectrometry (ICP-MS), we analyzed the concentrations of these metals and calculated bioconcentration factors (BCFs) to evaluate their transfer through trophic levels. Spatial analysis revealed that copper (up to 2528.20 mg/kg) and zinc (up to 1821.40 mg/kg) exceeded permissible limits, particularly near former mining sites. Elevated lead (807.59 mg/kg) and cadmium (2.94 mg/kg) were observed in industrial areas, while nickel and cobalt showed lower concentrations, but with localized peaks. Zinc was the most abundant metal in grass, while cadmium transferred efficiently to milk and cheese, raising potential health concerns. The results underscore the complex interplay between soil properties, contamination sources, and biological processes in heavy metal accumulation. These findings highlight the importance of continuous monitoring, risk assessment, and mitigation strategies to protect public health from potential exposure through contaminated dairy products.

## 1. Introduction

Minerals are vital dietary components for animals, serving a wide array of functions within the organism, including structural support, physiological processes, enzyme activity, and regulatory mechanisms [1,2]. Therefore, variations in the mineral content of both soil and feed directly influence the mineral status of animals, which in turn affects livestock productivity. Essential dietary elements are crucial for health and growth, while non-essential elements, often classified as potentially toxic, may enter the food chain without fulfilling any direct nutritional role [3]. Both mineral deficiencies and toxicities can adversely affect the health of humans and animals. Mild deficiencies in essential minerals present a particular challenge, as they can lead to detrimental health consequences without immediately displaying overt clinical symptoms. In contrast, severe deficiencies are typically more straightforward to diagnose and address, often manifesting in animals through symptoms such as infertility, abortions, poor weight gain, and anemia [4].

Milk and dairy products contain a spectrum of minerals [5]. Essential elements like iron, copper, and zinc are present in trace amounts. However, these trace elements can pose a greater risk in milk compared to other foods due to the high consumption rates in vulnerable populations (infants and the elderly), typically ranging from 30 to 150 kg/person/year [6]. Conversely, the presence of even low concentrations of non-essential or toxic elements like lead and cadmium can lead to serious health consequences [6,7,8,9]. Heavy metal exposure has been linked to kidney damage, genetic mutations, nervous system disorders, cardiovascular issues, various cancers, respiratory problems, immune system suppression, and infertility [5,10,11]. Therefore, strict monitoring of milk quality is crucial to minimize the potential health risks associated with heavy metal contamination.

Despite global sheep milk production reaching an estimated 10.62 million tonnes in 2020, its consumption in liquid form remains uncommon [12]. Consequently, the primary application for sheep milk lies in the diverse range of derived dairy products it yields [13]. These products encompass yogurt, ayran, butter, ice cream, kefir, cheese, and, importantly, sheep milk powder, which caters to the growing global demand for specialized dairy products [12]. Extensive research into the compositional characteristics of mammalian milk has consistently revealed that sheep milk stands out as the closest alternative to human breast milk [14]. This remarkable similarity stems from the comparable nutrient profiles, including essential amino acids, fatty acids, vitamins, and minerals, which are crucial for infant growth and development [14]. Consequently, sheep milk presents a promising and well-supported option for infant nutrition, particularly in instances where breastfeeding is not feasible or supplemented [14]. Sheep milk transcends its well-established nutritional value by offering a unique profile of bioactive components [14]. These bioactive ingredients, beyond basic nutrition, promote health benefits and position sheep milk as one of the world’s most important functional foods [14].

While milk and dairy products are a valuable source of essential nutrients like proteins, vitamins, lactose, unsaturated fatty acids, and minerals, they can also contain trace amounts of harmful pollutants [5]. Certain plants, such as *Oonopsis* and *Xylorrhiza*, can accumulate [15] selenium, potentially leading to livestock intoxication if consumed during grazing [5]. Additionally, contaminated soil from industrial activities or natural geological processes can introduce other toxins into the food chain, posing a potential human health risk [16]. Therefore, a delicate balance exists between the nutritional benefits of dairy and the need for stringent quality control to minimize exposure to contaminants [17].

Sheep milk composition can vary in its content of nutritionally important elements and potential toxins [18]. These variations depend on both intrinsic factors (lactation stage, breed, and animal health) and extrinsic factors (season, diet, and environment) [18]. Due to their grazing behavior, sheep can act as environmental bio-indicators [18]. Their milk serves as a valuable matrix for monitoring environmental pollution, particularly for heavy metals [16]. Well-known toxic heavy metals like arsenic, cadmium, lead, thallium, and mercury are ubiquitous environmental contaminants [8]. Grazing animals accumulate these metals through ingestion of contaminated water, plants, and feed [18].

Human exposure to heavy metals occurs through multiple pathways, including contaminated drinking water, dust inhalation, and plant uptake from polluted soil or groundwater [5]. The mobility and plant accumulation of these trace elements [19] are highly dependent on various factors such as soil type, micronutrient content, moisture, and pH [20]. This contamination can biomagnify through the food chain [5]. As livestock graze on contaminated pastures or consume concentrated feeds containing heavy metals, these elements can transfer to the animals, although the extent of the transfer can vary significantly, particularly in cattle [21,22,23]. Several studies highlight the transfer of heavy metals from contaminated environments to livestock. In one study, cows grazing near industrial areas in Greece accumulated higher copper levels in their livers compared to kidneys and muscles, with minimal transfer to milk [5,24]. Conversely, a study in Nigeria found lead exceeding safe limits in forage and cow’s milk, blood, and feces from animals grazing on lead-contaminated pastures [5,25]. These findings emphasize the variable transfer rates of heavy metals in animals. Additionally, research suggests that milk protein is particularly susceptible to heavy metal contamination, with cadmium preferentially binding to casein fractions in milk [5]. Furthermore, processing methods like adding coagulant and salt during cheese production can concentrate heavy metals in the final product compared to milk [5].

A holistic approach analyzing a broad spectrum of elements in various environmental matrices (soil, green grass, milk) is crucial to understand the transfer of minerals, including both essential and non-essential trace elements, along the soil–feed–animal continuum [26]. Blood mineral content, for example, directly influences milk composition [2]. Elevated heavy metal concentrations in blood can disrupt biological processes, leading to health problems and decreased productivity [27]. Furthermore, agricultural practices like pesticide and fertilizer use can alter the mineral profile of soil and plants. Due to their grazing behavior [28], sheep are particularly susceptible to bioaccumulating heavy metals from contaminated forage or water sources, potentially leading to their presence in milk [29]. Therefore, sheep can serve as valuable environmental bio-indicators of heavy metal contamination [2,18].

Although milk may not uniformly reflect the accumulation of all elements [18], it still provides valuable information within the broader context of mineral dynamics in pasture-based farming systems. Moreover, toxic levels of essential minerals and heavy metals can be found in milk, which may pose a risk for the consumer [30,31]. Since milk is a fundamental part of the human diet, controlling the levels of minerals and heavy metals in milk is a prime concern to protect human health [2]. In addition to milk, wool is another good bio-indicator for assessing environmental conditions, including heavy metal levels in soil, water, and air [2]. In many cases, wool proves to be a better bio-indicator than blood, urine, or animal milk, as it reflects feed quality and environmental [32,33].

Building upon previous research, the primary objective of this study was to comprehensively quantify the concentrations of micronutrients (^64^Cu, ^65^Zn), ultratrace elements (^52^Cr, ^59^Co, ^60^Ni), and heavy metals (^75^As, ^111^Cd, ^201^Hg, ^208^Pb) across various matrices, including soil, green grass, milk, cheese, and serum samples obtained from a flock of sheep. While existing studies have investigated the presence of these elements in environmental and food matrices, a significant knowledge gap persists regarding their distribution within specific agricultural contexts, particularly in areas impacted by mining activities. This study uniquely addresses this gap by focusing on samples collected from regions near the former Herja mine and the Aurul settling pond in Romania, where historical mining practices may have contributed to elevated levels of contamination. The selection of the indigenous Țurcana sheep breed, known for its adaptability to local conditions, further enhances the study’s significance. By investigating this breed, the study aims to provide valuable insights into the bioaccumulation of heavy metals within a specific agricultural framework, thereby contributing to the broader understanding of environmental health risks associated with mining. This study not only fills a crucial gap in the existing literature but also offers a novel perspective on the interactions between heavy metal contamination and agricultural practices in mining-affected areas. The findings aim to inform future monitoring and mitigation strategies, emphasizing the need for ongoing research into the effects of heavy metals on local ecosystems and food safety.

## 2. Materials and Methods

To isolate the effects of land use and grazing systems on mineral profiles, the study focused on farms with comparable soil and climatic conditions. Grazing areas were sampled between May and June 2024, coinciding with the western Mediterranean’s typical spring season. This period is characterized by mild temperatures and increasing daylight hours, alongside moderate precipitation. The investigated soils from Ferneziu and Firiza (Baia Mare, Romania) exhibit diverse pedological characteristics. In the northern and northeastern regions, regosols, eutricambosols, districambosols, and andosols developed on volcanic deposits are prevalent. Conversely, the southern region is dominated by aluvisols, luvisols, and stagnosols [34]. Within the upper Ilișua Valley watershed, the prevailing forested environment has shaped the soil types in Tîrlișua (Bistrița-Năsăud, Romania). Characterized by active humification of the organic surface layer, these soils are predominantly classified as brown, acidic brown, brown podzolic, and podzolic, reflecting the bioaccumulation processes that have occurred in this region.

### 2.1. Research Location

A total of 576 samples were collected across five categories: soil (144 samples, designated S_1-2024_–S_48-2024_); green grass (144 samples, G_1-2024_–G_48-2024_); sheep’s milk (144 samples, M_1-2024_–M_48-2024_); sheep’s cheese (144 samples, C_1-2024_–C_48-2024_); and sheep’s serum (144 samples, S_1-2024_–S_48-2024_). A total of 36 pre-determined locations were selected for sample collection across four areas: Baia Mare [Ferneziu I (8.0–11.5 km from the former Herja Mine); Ferneziu II (5.5–7.5 km from the former Herja Mine); Firiza (16.5–17.0 km from the Aurul settling pond in Tăuții de Sus)]; and Tîrlișua [area that served as a control site for establishing baseline levels of contaminants]. Three replicate samples were collected from each location. To establish a baseline for comparison and assess potential pollution, an additional control area, Tîrlișua, devoid of known heavy metal sources or prior related studies, was included. The spatial distribution of the sampling points is comprehensively illustrated in Figure 1. Precise geospatial data collection was ensured through the utilization of a handheld GPS device, the Garmin eTrex 32 × TopoActive Europa 2.2.

### 2.2. Collection of Soil, Green Grass, Sheep’s Milk, Cheese, and Sheep’s Serum Samples

#### 2.2.1. Surface Soil Sampling

Soil samples were collected from four locations using an opportunistic sampling method, as described by Bora et al., (2023) [16], during the spring-summer seasons of 2024. Topsoil samples (approximately 0.5 kg) were collected from each pasture sampling point using a composite approach. Three sub-samples were obtained from an area of approximately 100 × 100 m at each site. Sampling locations were chosen based on accessibility and with prior authorization from property owners. Soil samples were collected from the uppermost 10 cm, excluding large aggregates and debris, using a PVC corner and plastic trowel (ISO 11464/1994). Following collection, samples were transported to the laboratory, where roots and stones were removed, and the soil was homogenized through thorough mixing. Appendix A presents detailed information on the geographic origin, sampling depth, anthropogenic impact, and proximity of sheepfolds to pollution sources for the soil samples.

#### 2.2.2. Green Grass Sampling

The reason behind the collection of green grass samples probably stems from the understanding that sheep have a daily intake of about 1.2 kg of green grass per day. The sheep’s diet consisted exclusively of fresh, green grass, ensuring a natural and consistent intake of nutrients. This grazing-based approach aligns with the natural foraging behavior of sheep and promotes their overall health and well-being. The analysis of grass samples allows for an assessment of the nutritional content available to sheep, which can then be used to estimate their nutrient intake. To assess the spatial patterns of heavy metal contamination, grass samples were collected from four locations: Ferneziu (near the former Herja Mine, at two distances: 5.5–7.5 km and 8.0–11.5 km); Firiza (16.5–17.0 km from the Aurul settling pond in Tăuții de Sus); and a control area, Tîrlișua, with no known history of heavy metal sources or related studies (corresponding to the soil sampling sites). Three replicate samples were collected per location using clean polyethylene bags in the field. Upon arrival at the laboratory, samples were transferred to clean cellulose bags to minimize contamination from trace metals potentially present in tap water. They were then meticulously washed three times with deionized water to remove adhering particles. Following washing, samples were oven-dried at 70 °C for 48 h to remove moisture and facilitate accurate dry weight measurement. A comprehensive description of the pasture type and the constituent floristic species within the sampling area is provided in the Appendix A accompanying this article (Appendix A). Three grass species, *Agrostis capillaris* (*A. tenuis*) (dog grass—popular name), *Festuca rubra* (red meadow—popular name), *Poa pratensis* ssp. *Angustifolia* (firuța—popular name), were consistently present across all the sampling [35]. Appendix A provides a comprehensive overview of the grassland sampling sites, including geographic coordinates, anthropogenic impacts, and proximity to pollution sources.

#### 2.2.3. Sheep’s Milk Sampling

Raw milk samples were obtained through manual milking under hygienic conditions to minimize potential contamination and ensure sample integrity for subsequent heavy metal analysis. Milk sample collection was synchronized with the farmers’ milking routine, occurring in the evening upon the sheep’s return from grazing. This approach ensured representative samples reflecting the animals’ typical dietary intake patterns. Adhering to standard milk sample collection practices, the initial milk stream (foremilk) was discarded. This eliminates foremilk, which may contain higher concentrations of contaminants concentrated at the udder teat opening. Animals were milked twice daily by hand, both in the morning and evening, with an average daily yield of 500–800 mL per animal. Notably, the breed of sheep used remained consistent throughout the study period. To minimize container-derived metal leaching, all the milk samples were collected in pre-washed polyethylene containers. The washing process employed a 65% nitric acid solution. Following collection in pre-washed 125-mL polyethylene bottles, milk samples were immediately placed in a cooler with ice packs for transport to the laboratory. Upon arrival, samples were stored at −20 °C until further analysis. To minimize potential external contamination during the milk sample collection, a multi-pronged approach was implemented. Firstly, manual milking was employed, eliminating the risk of metal introduction from mechanical milking equipment. Secondly, the sample collector donned nitrile or latex gloves to reduce hand-borne contaminants. Thirdly, thorough udder sanitation procedures were implemented prior to milking to prevent external bacterial contamination. Finally, each sample was assigned a unique identifier corresponding to the village of origin, facilitating traceability. Appendix A provides detailed information regarding the milk samples.

#### 2.2.4. Sheep’s Cheese Sampling

Cheese samples originated from the same sheep milk used for the milk sample collection, reflecting the composition of milk typically sold in local markets. This locally produced cheese, most likely a traditional variety, was prepared using a simple method. The cheesemaking process began with curdling the milk. Approximately 22 L of fresh sheep’s milk, a representative sample from a flock of around 80 sheep in each studied area, was used. This approach ensured that the cheese reflected the average milk composition within each region of interest. Curdling was achieved by adding 8 mL of synthetic rennet (Cheag lichid (250 mL bottle)—a Romanian product was used) to the milk in a large plastic container. Synthetic rennet was specifically chosen as the ideal coagulant for this particular cheese type. This choice is likely due to its consistent performance and ease of use compared to traditional rennet derived from animal sources. The mixture was left undisturbed for about 25 min to allow curd formation. Following this holding period, the curds were gently mixed, and then separated from the whey using cheesecloth. This separation yielded approximately 4.5 kg of cheese and 16.8 L of whey. The cheese production environment was carefully controlled to ensure optimal curd formation and prevent undesired microbial growth. The room was maintained at a temperature of approximately 38 °C, which falls within the mesophilic temperature range commonly used for artisanal cheesemaking. This temperature range promotes the activity of mesophilic cheese cultures, responsible for the development of desirable flavors and textures. Furthermore, the humidity was kept below 60%. This relatively low humidity level discourages the growth of molds, which can negatively affect the cheese’s quality and safety. By controlling both temperature and humidity, the cheesemakers create an environment conducive to the production of a high-quality product with the intended characteristics.

For analysis, approximately 150 g of fresh sheep’s cheese were utilized. Each cheese sample was carefully weighed and then packaged in appropriately labeled plastic bags. This labeling ensured proper identification and tracking throughout the subsequent analysis process. Following collection, the cheese samples were subjected to controlled storage conditions to minimize potential degradation before analysis. The majority of the samples were refrigerated at a temperature range of 2–4 °C. This temperature range effectively inhibits the growth of most spoilage microorganisms while preserving the cheese’s quality. However, in some instances, samples were stored at −18 °C in a freezer. This deeper freezing may have been necessary for specific analytical techniques or for extended storage durations. Appendix A provides detailed information regarding cheese samples.

#### 2.2.5. Sheep’s Serum Sampling

The research protocol adhered to the ethical guidelines established by the European Commission Directive 86/609/EEC concerning the welfare of animals used for experimental and other scientific purposes [36]. This on-farm experiment was conducted at a private dairy farm specializing in sheep milk and cheese production, situated near Baia Mare and Tîrlișua, Romania. Serum samples were obtained from sheep selectively chosen through a randomized process. These sheep originated from two distinct geographical locations: (I) areas with documented scientific evidence of heavy metal pollution (Ferneziu I–II and Firiza), and (II) a designated control area (Tîrlișua). Blood samples were collected from the sheep’s jugular vein in June using evacuated tubes devoid of anticoagulant, ensuring maintenance of the cold chain. Following collection, the blood samples were transported to the laboratory for further processing. Centrifugation at 3000 rpm for 15 min facilitated the separation of the serum from whole blood. The isolated serum samples were then transferred into 1.5 mL Eppendorf tubes and stored at −26 °C in a freezer until analysis. Samples were secured in an upright position in appropriately sized shipping boxes using racks to minimize the risk of tipping or spillage during transport. Transportation protocols must prioritize maintaining sample integrity by minimizing the risk of tipping or spillage. Sample transport ensured delivery to the laboratory within a 72-h timeframe. Appendix A provides detailed information regarding the sheep’s serum samples. Appendix A provides detailed information regarding the sheep population under investigation.

### 2.3. Sample Preparation and Digestion Using Microwave Digestion System

#### 2.3.1. Soil Sample Preparation

Topsoil samples underwent oven drying at 105 °C for a minimum of 72 h in a Binder FD 53 oven (Darmstadt, Germany). Subsequently, the samples were sieved and disaggregated by passing them through a 2-mm nylon mesh using a Retsch 110 automatic mill (Darmstadt, Germany). The dried and sieved topsoil underwent homogenization, followed by subsampling. The sub-samples were then manually pulverized using an agate mortar and pestle. The soil sample preparation procedure employed in this study adheres to a well-established method documented in previous research [16,37,38]. This study adopted a microwave digestion method for soil samples previously optimized using a Milestone START D Microwave Digestion System (Sorisole, Italy) [16,37,38]. Briefly, 0.2–0.5 g of dried and milled soil was weighed directly into a clean Teflon digestion vessel. Subsequently, 12 mL of aqua regia (composed of 9 mL HCl and 3 mL HNO_3_) was added to the vessel. Following a 15-min incubation period, the mineralization process was conducted using the Milestone START D System. The digestion was carried out with the program described in Appendix A. Following complete sample mineralization, the Teflon digestion vessel was carefully opened under strict safety protocols. The resulting solution was then filtered through a 0.45-μm PTFE membrane filter to remove any undissolved particles. The filtrate was quantitatively transferred to a 50-mL volumetric flask and brought to volume with deionized water after cooling to room temperature.

#### 2.3.2. Green Grass Sample Preparation

Green grass samples were meticulously rinsed with tap water to eliminate adhering dust, soil particles, and potential organisms. Subsequently, the samples were thoroughly rinsed with deionized water. The washing duration varied from 5 to 10 min at 30 °C, dependent on the sample size and quantity. The ground samples were then packed in a Ziplock bag and kept in the refrigerator under −4 °C prior. Green grass sample preparation and digestion adopted previously described methods by Bora et al., (2020) [38] and Bora et al., (2022) [39], with some modifications. Only the edible portions of each grass type were utilized for analysis. Prior to the sample preparation, any damaged or spoiled sections of the green grass samples were discarded. Fresh samples were weighed accurately and subsequently sectioned into diminutive proportions using a pre-cleaned scalpel for oven drying at 105 °C. A Binder FD 53 oven (Darmstadt, Germany) was employed for this drying process. The drying process was determined to be complete upon achieving a constant sample weight. This typically required a drying time of 72 to 92 h. Following drying, the samples were pulverized using a Retsch 110 automatic mill (Darmstadt, Germany) and sieved through a 2-mm mesh to obtain a homogenous powder with a particle size less than 2 mm. Following drying and grinding, the samples were transferred to pre-labeled polyethylene containers for storage until further digestion. This study adopted a previously optimized microwave digestion method for sample preparation, as described by Bora et al., (2015) [37] and Bora et al., (2022) [39]. Briefly, 0.5 g of dried, pre-chopped green grass sample (prepared using a mortar and pestle followed by blending) was weighed accurately and transferred to a pre-cleaned Teflon digestion vessel. Subsequently, 7 mL of 65% nitric acid (HNO_3_) and 2 mL of hydrogen peroxide (H_2_O_2_) were added to the vessel. The vessel was then securely sealed and placed within a Milestone START D Microwave Digestion System (Sorisole, Italy). The microwave digestion program employed is detailed in Appendix A. Following digestion, the samples were subjected to filtration using Whatman 42 filter paper into a pre-cleaned 10-mL volumetric flask. The filter paper was then rinsed with deionized water to ensure complete transfer of the filtrate. The combined filtrate and rinsing solution were used to achieve the final volume by diluting to the mark with deionized water.

#### 2.3.3. Sheep’s Milks and Cheese Sample Preparation

Ovine milk samples were collected directly from the udder during the morning milking session at peak yield. Fifty-milliliter Falcon tubes (Fisher Scientific, Waltham, MA, USA) were pre-cleaned for sample collection by washing them with a 10% nitric acid (HNO_3_) solution and subsequently rinsing them thoroughly with deionized water to eliminate any residual acid. The collected milk samples were then transferred to these pre-washed tubes and stored frozen at −60 °C (NordicLab ULT U100) until further processing and analysis. The determination of heavy metals in milk samples necessitated a digestion procedure adhering to the guidelines established in UNI EN 13805:2002 [40]. Aliquots of approximately 0.5 mL milk were transferred into pre-decontaminated PTFE-TFM (polytetrafluoroethylene-tetrafluoroethylene) vessels. The vessel pre-treatment involved decontamination with 3 mL of ultrapure nitric acid (60% *v*/*v*) followed by 5 mL of ultrapure water. Milk sample digestion employed a modified version of AOAC method 986.15 [41]. The initial step involved transferring a 0.5 mL aliquot of the milk sample into a pre-decontaminated PTFE-TFM (polytetrafluoroethylene-tetrafluoroethylene) vessel. Subsequently, a pre-mixed solution consisting of 9.00 mL of 65% HNO_3_ and 1.00 mL of 72% HClO_4_ was added to the digestion vessel. Following secure closure, the vessel was positioned within a Milestone START D Microwave Digestion System (Sorisole, Italy). The specific microwave digestion program utilized is detailed in Appendix A for reference. Following microwave digestion, the samples were filtered through Whatman 42 filter paper into a pre-cleaned 10 mL volumetric flask. To ensure complete transfer of the analytes, the filter paper was rinsed with deionized water. The collected filtrate and rinsing solution were then combined within the volumetric flask, and the final volume was achieved by diluting to the mark with deionized water.

Cheese samples were procured from the same farms that provided the milk samples. All the cheese samples were homemade products, typically sold within local markets, and underwent minimal industrial processing. Whole cheese samples were obtained directly from producers and immediately transported to the laboratory under refrigerated conditions (4 °C) within a 3–5-h timeframe. Sample analysis commenced within several days of laboratory arrival. The cheese samples were homogenized through grating and subsequently divided into three plastic containers for subsequent heavy metal analysis. To facilitate mineralization, 7 mL of concentrated nitric acid (HNO_3_, 65%) and 2 mL of hydrogen peroxide (H_2_O_2_) were added to the digestion vessel. The sealed vessel was subsequently subjected to microwave-assisted digestion using a Milestone START D system (Sorisole, Italy), with the specific parameters outlined in Appendix A. Post-digestion, the sample solution was filtered through Whatman 42 filter paper into a pre-cleaned 10 mL volumetric flask. Quantitative recovery of the filtrate was ensured by rinsing the filter paper with deionized water, followed by dilution to the final volume with the same solvent.

#### 2.3.4. Sheep Serum Sample Preparation

Serum samples were digested using the wet acid digestion method with slight modification. To each sample, 300 μL of concentrated nitric acid (HNO_3_, Ultrex, Fisher), 200 μL of concentrated hydrochloric acid (HCl, Ultrex, Fisher), and 100 μL of 30% hydrogen peroxide (H_2_O_2_, Ultrex, Fisher) were added. The total volume was adjusted to 2.0 mL with deionized water. An acid-washed stir bar was included in each vial to facilitate digestion during the microwave heating process, following the parameters outlined in Appendix A. Upon completion of the microwave digestion process, the samples were removed and allowed to cool to ambient temperature. Subsequently, the digests were quantitatively transferred to pre-cleaned 15-mL polypropylene tubes. The samples were then diluted to the desired volume using deionized (DI) water and stored at a controlled temperature of 8 °C until analysis.

### 2.4. General ICP-MS Instrumental Parameters of Analysis

Micronutrient (copper-64 [^64^Cu], zinc-65 [^65^Zn]), ultratrace element (chromium-52 [^52^Cr], cobalt-59 [^59^Co], nickel-60 [^60^Ni]), and heavy metal (arsenic-75 [^75^As], cadmium-111 [^111^Cd], mercury-201 [^201^Hg], lead-208 [^208^Pb]) concentrations were quantified using inductively coupled plasma mass spectrometry (ICP-MS). The instrument employed was an iCAP Q ICP-MS (Thermo Fisher Scientific, Waltham, MA, USA) equipped with an ASX-520 autosampler, a micro-concentric nebulizer, Ni sampler and skimmer cones, and a peristaltic sample delivery pump. Quantitative analysis mode was utilized for data acquisition. Sample introduction into the ICP-MS plasma was facilitated by a nebulizer connected to a cyclonic spray chamber. The standard ICP-MS torch featured a 1.5-mm diameter injector. Notably, collision cell technology, a crucial component of the ICP-MS, efficiently eliminated common interferences through the use of pure helium as the collision cell gas and the application of kinetic energy discrimination (KED) mode, enabling the detection of multiple elements with minimal interference.

Prior to quantitative analysis, the ICP-MS underwent a stabilization period of at least 45 min following startup. During this time, experimental conditions were verified, and mass calibration was performed. A short-term stability test was conducted using a tuning standard solution (TUNE B iCAP Q) containing Ba, Bi, Ce, Co, In, Li, and U (each at 1.0 μg/L) in a 2% HNO_3_ + 0.5% HCl matrix. This auto-tuning process optimized the plasma sampling zone to achieve a balance between high sensitivity, optimized ion optics voltages, and minimal formation of cluster and doubly charged ions. Daily optimization ensured maximum sensitivity for M^+^ ions. Double ionization and oxide formation were monitored using the ratios of Ba^2+^/Ba^+^ and Ce^2+^/CeO^+^, respectively, ensuring these values remained below 2%. Argon (Ar 5.0) and helium (He 6.0) carrier gases were employed with a purity of 99.99% (Messer, Gumpoldskirchen, Austria). All the samples were analyzed in triplication, with each analysis consisting of seven replicates. Detailed instrumental parameters for ICP-MS analysis have been previously reported [16,38]. The operating parameters specific to this study are provided in Appendix A.

### 2.5. Reagents and Equipment

All the chemicals and reagents employed were of high purity, sourced from either Merck or Sigma-Aldrich (Darmstadt, Germany). These included ultrapure nitric acid (HNO_3_, 65%) for trace analysis, ultrapure hydrogen peroxide (H_2_O_2_) for trace analysis, and high-purity deionized water (18.2 MΩ⋅cm^−1^ resistivity) obtained from a Milli-Q Integral Ultrapure Water System (Type 1). Teflon digestion vessels were cleaned with 25 mL HNO_3_ before each mineralization step. The mineralization process of soil, green grass, sheep’s milk, cheese, and serum samples was performed in triplicate, with a maximum of six digestion vessels per run (five for samples and one for the blank). These vessels were constructed from modified polytetrafluoroethylene (TFM-PTEE). All the flasks utilized in the experiment were pre-treated with 5 M HNO_3_ for 24 h and subsequently rinsed thoroughly with deionized water (3–4 repetitions). A high-precision analytical balance (KERN ADB 100-4) (Ebingen, Germany) was employed for weighing both soil and honey samples, as well as for the preparation of working and calibration solutions.

### 2.6. Quality Control of the Chemical Analyses

Following the guidelines outlined in Commission Regulation (EU) No. 2016/582 [42], limits of detection (LoDs) and quantification (LoQs) were established for the analyzed elements. These values were determined using the standard deviation (σ) of blank solution measurements (n = 20). To ensure analytical accuracy, a multi-element internal standard solution containing indium, scandium, and praseodymium at a final concentration of 10 ng/mL was added. Specifically, LoD was calculated as 3σ and LoQ as 10σ. Repeatability was assessed using the Horwitz Ratio (HorRat), calculated by dividing the measured relative standard deviation (RSDr) by the value estimated from the Horwitz equation [43]. All the HorRat values were confirmed to be less than 2. Detailed validation parameters, including precision, accuracy, recovery, and uncertainty, are presented in Appendix A. Calibration standards were prepared using ICP Multi-Element Standard Solution XXI CertiPUR, encompassing a range of five concentrations (2.5, 5, 10, 25, and 50 µL). Both the accuracy and precision of the analytical procedure were evaluated by spiking a known amount of the target metal into a sample aliquot and analyzing it alongside the original sample. Precision was expressed as the percent relative standard deviation (RSD%) of triplicate analyses. Recovery assays were performed on spiked soil, green grass, sheep’s milk, cheese, and serum samples at a concentration of 50 µL (n = 3 replicates). The average recovery (R%) achieved in this experiment ranged from 91.5% to 117.89%.

### 2.7. Statistical Analysis

Data acquisition and basic descriptive statistics (mean, median, relative standard deviation) were performed using Microsoft Excel 365 (Microsoft Corporation, Redmond, WA, USA) and Addinsoft version 15.5.03.3707 (Addinsoft, a subsidiary of Microsoft). Data precision was assessed and expressed as standard deviation (SD). All the data are presented as means ± standard deviations. Statistical analysis was conducted using IBM SPSS Statistics version 24 (IBM Corp., Armonk, NY, USA). Replicate measurements (n = 3) were averaged and presented alongside their standard deviations. A two-way analysis of variance (ANOVA) was employed to assess the influence of different variables on the concentrations of heavy metals in soil, green grass, sheep milk, cheese, and serum. The statistical software package SPSS Version 24 was utilized for both the ANOVA test and subsequent mean separation using Duncan’s multiple range test. A significance level of α ≤ 0.005 was adopted.

## 3. Results and Discussion

### 3.1. Soil

A total of 144 soil samples were collected from a depth of 0–10 cm across various locations within the Baia Mare (Ferneziu and Firiza) and Tîrlișua (Bistrița-Năsăud) region. The samples were subsequently analyzed to determine the concentration of nine elements commonly associated with heavy metal pollution: copper (Cu), zinc (Zn), lead (Pb), cadmium (Cd), nickel (Ni), cobalt (Co), arsenic (As), chromium (Cr), and mercury (Hg). The results are presented in Table 1. This study aimed to assess the extent of heavy metal contamination specifically in the Ferneziu and Firiza areas, utilizing Tîrlișua as a control location. Additional information regarding the maximum and minimum concentration of the analyzed metals depending on the area is presented in Appendix A.

Analysis of heavy metal concentrations in the soil samples revealed significant variations across the study area. Maximum concentrations at a depth of 0–10 cm were observed for Cu (1072.35 mg/kg) followed by Zn (852.78 mg/kg), Pb (137.73 mg/kg), Cr (4.58 mg/kg), Ni (1.44 mg/kg), Cd (1.14 mg/kg), Co (0.36 mg/kg), As (0.35 mg/kg), and Hg (BDL ≥ LoQ for Hg = 0.1379 µg/L). Median concentrations across the entire sampling area followed a similar trend: Zn (1041.26 mg/kg) > Cu (322.82 mg/kg) > Pb (243.50 mg/kg) > Cr (5.47 mg/kg) > Ni (1.96 mg/kg) > Cd (1.62 mg/kg) > Co (0.76 mg/kg) > As (0.67 mg/kg) > Hg (BDL). Notably, the concentrations of several heavy metals in the soil samples surrounding the sheepfold significantly exceeded the established regulatory limits. These limits include the Romanian Regulation of allowable quantities of hazardous and harmful substances in soil (Order of the Ministry of Waters, Forest and Environmental Protection No. 765/3 November 1997) and the Council Directive 86/278/EEC for Protection of the Environment (European Communities Council 1986) [44].

Copper (Cu) exhibited significant spatial variations in the soil samples (Appendix A), with concentrations ranging from a minimum of 0.89 mg/kg to a maximum of 2528.20 mg/kg (average value: 1072.35 mg/kg). The highest concentrations were observed in Ferneziu, located closest (6–7 km) to the former Herja mine, the identified source of pollution (322.82 mg/kg). Cu concentrations decreased with increasing distance (285.81 mg/kg at 10–12 km and 246.50 mg/kg at 16.5 km, furthest from the mine). Similarly, Firiza displayed elevated Cu (808.72 ± 134.94 mg/kg) compared to the control area (Tîrlișua: 0.89 ± 0.33 mg/kg). These findings suggest potential environmental risks associated with Cu contamination, particularly near the mine and in Firiza.

The observed Cu concentrations (0.89–2528.20 mg/kg) were comparable to findings reported in various studies on heavy metal-polluted soils in Romania. These include investigations near Baia Mare (19.8–4155.94 mg/kg) [16,37,45,46,47], Copșa Mică (77–7675 mg/kg) [48], and Huși (256.00 mg/kg) [45]. Notably, some studies reported Cu concentrations exceeding ours, particularly in the Baia Mare area [16,37,38]. In contrast, Cu levels in vineyard soils from Caraș-Severin (36.63–112.00 mg/kg) [49] were generally lower. Notably, the observed Cu concentrations in all areas except the control site exceeded the established maximum permissible limits.

Zinc (Zn) concentrations exhibited a wide range, spanning from a minimum of 24.66 mg/kg to a maximum of 1821.40 mg/kg (Appendix A). The average Zn concentration across the study area was 852.78 mg/kg. Zn concentrations exhibited a distinct spatial pattern, with the highest levels recorded in Ferneziu (1821.40 mg/kg), located 7.5 km from the pollution source (Table 1). This was followed by Zn concentrations in the sample collection area at 5 km (1544.99–1602.17 mg/kg). Zinc (Zn) concentrations exhibited a distinct spatial pattern, with the highest levels recorded in Ferneziu (1041.26 mg/kg), located closest (6–7 km) to the identified pollution source, the former Herja mine. Zn concentrations decreased with increasing distance from the mine, as evidenced by 877.45 mg/kg at 16.5 km (the furthest location) and 708.66 mg/kg at 10–12 km. Consistent with the observed pattern for copper (Cu), zinc (Zn) concentrations also exhibited a significant inverse relationship with distance from the pollution source. This finding further supports the notion that the identified pollution source is the primary contributor to elevated Zn levels in the surrounding environment. The observed Zn concentrations align with the findings of previous studies in the region. Huzum et al., (2012) [45] reported Zn concentrations of 61.10 mg/kg, while Bora et al., (2020) [38] found a wider range of 45.36–3483.25 mg/kg. Chakraborty et al., (2017) [46] and Mihali et al., (2017) [47] also reported Zn concentrations within the ranges of 54.4–2370.0 mg/kg and 82.26–1002 mg/kg, respectively. Notably, the Zn concentrations presented by Bora et al., 2015 [37] (69.44 mg/kg) are significantly lower than those observed in this study. Remarkably, Zn concentrations in all the investigated areas, excluding the control site, surpassed the established maximum permissible limits. This finding highlights the potential environmental risks associated with Zn contamination in these regions.

Soil Pb concentrations were highest in Ferneziu, with the closest samples (approximately 5.5 km) to the pollution source exhibiting the greatest levels (807.59 mg/kg and 694.50 mg/kg). A similar spatial trend was observed for Cd, with the highest concentrations (2.94–2.30 mg/kg) found in Ferneziu soil, but at a distance of 11.5 km from the former Herja mine. These findings suggest that proximity to pollution sources and potential accumulation processes influence heavy metal distribution in Baia Mare area soils. Notably, Ferneziu is known for mining and industrial activities, suggesting direct emissions as a key contributor. Furthermore, distance-dependent decreases in metal concentrations support this notion. Specific soil properties in Ferneziu, such as pH and organic matter content, might also play a role in metal mobility and retention, potentially explaining the observed accumulation pattern. Further investigation into these soil characteristics, along with vegetation cover and historical land use, could provide a more comprehensive understanding of the mechanisms governing heavy metal fate in the study area.

The results obtained for Pb and Cd were comparable to those obtained by Huzum et al., (2012) [45] (12.90 mg/kg Pb, 0.21 mg/kg Cd), Bora et al., (2020) [38] (6.62–4262.23 mg/kg Pb, 0.12–32.53 mg/kg Cd), Chakraborty et al., (2017) [46] (38.0–14,329.0 mg/kg Pb), Paulette et al., (2015) [48] (705–10.074 mg/kg Pb), Mihali et al., (2017) [47] (48.12–3472 mg/kg Pb, 0.04–11 mg/kg Cd), Albulescu et al., (2009) [49] (21.90 mg/kg Pb, 1.77 mg/kg Cd), Bora et al., 2015 [37] (14.77 mg/kg Pb, 0.45 mg/kg Cd), and Bora et al., 2023 [16] (1205.57 mg/kg Pb, 6.33 mg/kg Cd).

**Table 1 toxics-12-00752-t001:** The content of heavy metals in soil from Ferneziu, Firiza (Maramureș area) and Tîrlișua (Bistrița-Năsăud), Romania. (Mean ± standard deviation) (n = 3).

**Areas** **Sample Code** **Year of Harvest**	**Distance from the Source of Pollution (** **~) km**	**Sampling** **Depth** **(0–10 cm)**	**Cu** M.P.L. *	**Zn** M.P.L. *	**Pb** M.P.L. *	**Cd** M.P.L. *	**Ni** M.P.L. *	**Co** M.P.L. *	**As** M.P.L. *	**Cr** M.P.L. *	**Hg** M.P.L. *
			20 mg/kg	100 mg/kg	20 mg/kg	1 mg/kg	20 mg/kg	15 mg/kg	5 mg/kg	30 mg/kg	0.1 mg/kg
Alert threshold		Susceptible	100 mg/kg	300 mg/kg	50 mg/kg	3 mg/kg	75 mg/kg	30 mg/kg	15 mg/kg	100 mg/kg	1 mg/kg
		Less susceptible	250 mg/kg	700 mg/kg	250 mg/kg	5 mg/kg	200 mg/kg	100 mg/kg	25 mg/kg	300 mg/kg	4 mg/kg
Intervention threshold		Susceptible	200 mg/kg	600 mg/kg	100 mg/kg	5 mg/kg	150 mg/kg	50 mg/kg	25 mg/kg	300 mg/kg	2 mg/kg
		Less susceptible	500 mg/kg	1.500 mg/kg	1.000 mg/kg	10 mg/kg	500 mg/kg	250 mg/kg	50 mg/kg	600 mg/kg	10 mg/kg
Soil samples exposed to anthropogenic sources of heavy metals pollution											
Ferneziu	Near (~10/12 km) to the Herja Mine in Ferneziu										
S_1-2024_ 2024	The sheepfold was located approximately (~) 8.0 km from the former Herja mine in Ferneziu										
			2262.79 ± 253.90 ^ab^	244.05 ± 84.29 ^îj^	30.79 ± 18.71 ^kl^	1.57 ± 0.64 ^cdefghi^	BLD	2.13 ± 0.83 ^a^	0.69 ± 0.22 ^cdefg^	6.09 ± 0.91 ^abcdefg^	BLD
S_2-2024_ 2024			1854.71 ± 100.75 ^de^	1041.70 ± 189.43 ^de^	15.55 ± 6.94 ^l^	1.32 ± 0.81 ^defghiî^	0.60 ± 0.27 ^fgh^	1.06 ± 0.18 ^cd^	1.41 ± 0.15 ^ab^	7.24 ± 4.32 ^abcde^	BLD
S_3-2024_ 2024			2168.79 ± 299.09 ^bc^	1039.75 ± 174.27 ^de^	6.45 ± 4.17 ^l^	BLD	3.22 ± 0.57 ^bcd^	BLD	0.53 ± 0.12 ^defg^	3.33 ± 2.08 ^efghi^	BLD
S_4-2024_ 2024			2041.20 ± 85.82 ^bcd^	863.36 ± 84.31 ^defgh^	118.38 ± 115.89 ^fghiîj^	0.63 ± 0.13 ^iîj^	1.38 ± 0.15 ^efgh^	BLD	0.57 ± 0.39 ^defg^	6.33 ± 1.15 ^abcdef^	BLD
S_5-2024_ 2024			1772.44 ± 251.07 ^de^	625.14 ± 53.21 ^efghiî^	123.86 ± 28.39 ^fghi^	1.03 ± 0.51 ^fghiîj^	3.87 ± 0.87 ^bc^	BLD	0.30 ± 0.23 ^fgh^	9.60 ± 2.26 ^ab^	BLD
S_6-2024_ 2024	The sheepfold was located approximately (~) 11.5 km from the former Herja mine in Ferneziu										
			1789.68 ± 265.87 ^de^	1576.81 ± 210.27 ^ab^	436.36 ± 35.30 ^c^	2.57 ± 0.86 ^ab^	4.90 ± 0.30 ^b^	0.75 ± 0.43 ^de^	1.73 ± 0.72 ^a^	6.32 ± 1.19 ^abcdef^	BLD
S_7-2024_ 2024			1251.56 ± 113.27 ^fg^	797.19 ± 195.38 ^efghi^	32.15 ± 19.04 ^jkl^	0.58 ± 0.40 ^iîj^	0.80 ± 0.22 ^fgh^	BLD	0.93 ± 0.34 ^cd^	6.37 ± 4.58 ^abcdef^	BLD
S_8-2024_ 2024			1952.37 ± 68.13 ^bcde^	431.37 ± 102.48 ^hiîjg^	84.24 ± 24.99 ^ghiîjkl^	1.52 ± 0.39 ^cdefghi^	0.95 ± 0.61 ^fgh^	1.54 ± 0.63 ^bc^	BLD	2.24 ± 1.04 ^ghi^	BLD
S_9-2024_ 2024			1938.18 ± 73.35 ^cde^	291.87 ± 159.83 ^iîj^	110.64 ± 7.51 ^fghiîjk^	0.87 ± 0.35 ^hiîj^	1.76 ± 1.47 ^defgh^	0.51 ± 0.39 ^ef^	0.20 ± 0.07 ^gh^	8.68 ± 2.97 ^abc^	BLD
S_10-2024_ 2024			1323.72 ± 126.63 ^f^	754.50 ± 95.87 ^efghi^	61.05 ± 21.70 ^hiîjkl^	2.38 ± 0.32 ^abc^	2.93 ± 0.30 ^cde^	0.41 ± 0.26 ^ef^	0.52 ± 0.39 ^defg^	4.13 ± 1.27 ^efghi^	BLD
S_11-2024_ 2024			2194.87 ± 236.06 ^bc^	372.04 ± 37.27 ^hiîj^	416.94 ± 139.99 ^c^	0.55 ± 0.35 ^iîj^	2.31 ± 1.80 ^cdefg^	1.63 ± 0.47 ^ab^	0.44 ± 0.48 ^efgh^	1.83 ± 0.24 ^hi^	BLD
S_12-2024_ 2024			2528.20 ± 424.91 ^a^	476.11 ± 114.21 ^fghiîj^	18.96 ± 2.89 ^l^	2.60 ± 0.61 ^ab^	0.54 ± 0.45 ^hg^	1.11 ± 0.14 ^cd^	0.68 ± 0.52 ^cdefg^	3.47 ± 1.06 ^efghi^	BLD
Ferneziu	Near (~6/7 km) to the Herja Mine in Ferneziu										
S_13-2024_ 2024	The sheepfold was located approximately (~) 5.5 km from the former Herja mine in Ferneziu										
			1314.48 ± 139.42 ^f^	1544.99 ± 198.87 ^abc^	50.62 ± 15.01 ^iîjkl^	2.47 ± 0.71 ^abc^	9.02 ± 3.99 ^a^	0.30 ± 0.22 ^ef^	0.24 ± 0.18 ^gh^	6.66 ± 0.69 ^abcdef^	BLD
S_14-2024_ 2024			673.86 ± 123.08 ^îj^	767.73 ± 576.69 ^efghi^	159.56 ± 36.92 ^efg^	1.38 ± 0.46 ^defghiî^	4.85 ± 1.81 ^b^	0.68 ± 0.49 ^de^	BLD	1.57 ± 0.28 ^i^	BLD
S_15-2024_ 2024			1328.66 ± 38.46 ^f^	1602.17 ± 533.42 ^ab^	115.32 ± 27.32 ^fghiîj^	2.01 ± 1.43 ^abcdef^	0.32 ± 0.20 ^h^	BLD	BLD	2.26 ± 1.25 ^fghi^	BLD
S_16-2024_ 2024			1081.10 ± 225.24 ^fghi^	711.15 ± 151.80 ^efghiî^	55.65 ± 23.47 ^hiîjkl^	1.68 ± 0.86 ^abcdefgh^	BLD	BLD	BLD	8.21 ± 2.57 ^abcd^	BLD
S_17-2024_ 2024			1154.75 ± 170.18 ^fgh^	768.42 ± 37.11 ^efghi^	694.50 ± 48.11 ^b^	2.94 ± 0.37 ^a^	BLD	BLD	0.86 ± 0.36 ^cde^	2.04 ± 1.12 ^hi^	BLD
S_18-2024_ 2024			925.87 ± 44.51 ^hiî^	698.23 ± 39.42 ^efghiî^	807.59 ± 90.17 ^a^	1.27 ± 0.29 ^defghiî^	2.32 ± 1.75 ^cdefg^	BLD	0.30 ± 0.22 ^fgh^	4.89 ± 1.95 ^cdefghi^	BLD
S_19-2024_ 2024	The sheepfold was located approximately (~) 7.5 km to the former Herja mine in Ferneziu										
			2186.70 ± 332.47 ^bc^	1821.40 ± 40.92 ^a^	311.67 ± 40.06 ^d^	1.98 ± 1.37 ^abcdefg^	1.36 ± 0.85 ^efgh^	0.76 ± 0.41 ^de^	0.49 ± 0.37 ^defgh^	5.43 ± 3.58 ^cdefghi^	BLD
S_20-2024_ 2024			1002.16 ± 114.60 ^ghiî^	1325.17 ± 985.25 ^abc^	172.25 ± 52.36 ^ef^	0.81 ± 0.23 ^hiîj^	0.80 ± 0.35 ^fgh^	0.74 ± 0.44 ^de^	0.26 ± 0.24 ^gh^	5.73 ± 2.06 ^cdefgh^	BLD
S_21-2024_ 2024			638.79 ± 162.34 ^îj^	881.29 ± 99.96 ^defg^	121.65 ± 15.93 ^fghiî^	0.98 ± 0.08 ^ghiîj^	BLD	BLD	0.76 ± 0.45 ^cdef^	3.73 ± 2.51 ^efghi^	BLD
S_22-2024_ 2024			1676.33 ± 62.28 ^e^	1044.22 ± 87.64 ^de^	48.27 ± 22.71 ^iîjkl^	0.94 ± 0.35 ^hiîj^	1.14 ± 0.18 ^efgh^	0.78 ± 0.67 ^de^	0.38 ± 0.22 ^efgh^	10.01 ± 2.36 ^a^	BLD
S_23-2024_ 2024			885.51 ± 182.79 ^hiî^	357.77 ± 37.07 ^iîj^	179.80 ± 52.85 ^ef^	2.30 ± 0.27 ^abcd^	0.74 ± 0.52 ^fgh^	0.51 ± 0.30 ^de^	1.08 ± 0.18 ^bc^	6.72 ± 0.41 ^abcdef^	BLD
S_24-2024_ 2024			857.21 ± 239.96 ^hiî^	972.54 ± 140.55 ^def^	205.11 ± 15.63 ^e^	0.68 ± 0.38 ^hiîj^	1.19 ± 0.28 ^efgh^	0.30 ± 0.22 ^ef^	0.61 ± 0.25 ^defg^	2.80 ± 0.99 ^fghi^	BLD
Firiza	Near (~17 km) to the Aurul settling pond mining (decant pond) in Tăuții de Sus										
S_25-2024_ 2024	The sheepfold was located approximately (~) 16.5 km from the Aurul settling pond mining (decant pond) in Tăuții de Sus										
			324.73 ± 27.95 ^klm^	873.64 ± 131.98 ^defg^	37.22 ± 17.70 ^jkl^	0.34 ± 0.28 ^îj^	0.50 ± 0.34 ^gh^	BLD	BLD	4.69 ± 2.40 ^cdefghi^	BLD
S_26-2024_ 2024			459.97 ± 156.09 ^jk^	877.39 ± 80.96 ^defg^	53.54 ± 14.73 ^hiîjkl^	BLD	0.28 ± 0.26 ^h^	BLD	BLD	2.99 ± 0.65 ^fghi^	BLD
S_27-2024_ 2024			143.72 ± 40.95 ^lm^	697.47 ± 297.79 ^efghiî^	45.69 ± 25.10 ^iîjkl^	0.63 ± 0.23 ^iîj^	0.98 ± 0.20 ^fgh^	BLD	BLD	3.38 ± 0.91 ^efghi^	BLD
S_28-2024_ 2024			88.62 ± 14.12 ^m^	255.70 ± 93.89 ^îj^	75.83 ± 55.83 ^hiîjkl^	0.81 ± 0.23 ^hiîj^	BLD	BLD	BLD	1.87 ± 0.55 ^hi^	BLD
S_29-2024_ 2024			808.72 ± 134.94 ^iî^	1339.72 ± 231.08 ^abc^	135.32 ± 42.64 ^efgh^	1.21 ± 0.64 ^defghiî^	BLD	BLD	BLD	2.05 ± 1.82 ^hi^	BLD
S_30-2024_ 2024			116.45 ± 77.41 ^m^	421.83 ± 223.35 ^hiîjg^	65.19 ± 37.79 ^hiîjkl^	BLD	1.04 ± 0.21 ^fgh^	BLD	BLD	6.56 ± 2.12 ^abcdef^	BLD
S_31-2024_ 2024			23.60 ± 10.63 ^m^	1119.87 ± 119.59 ^cde^	18.50 ± 11.05 ^l^	BLD	0.65 ± 0.16 ^fgh^	BLD	BLD	2.16 ± 0.24 ^ghi^	BLD
S_32-2024_ 2024			435.79 ± 191.44 ^jkl^	1589.19 ± 375.79 ^ab^	69.89 ± 6.58 ^hiîjkl^	1.05 ± 0.17 ^fghiî^	2.48 ± 1.33 ^cdef^	BLD	BLD	5.02 ± 3.30 ^cdefghi^	BLD
S_33-2024_ 2024	The sheepfold was located approximately (~) 17.0 km from the Aurul settling pond mining (decant pond) in Tăuții de Sus										
			220.79 ± 127.27 ^klm^	995.16 ± 124.57 ^de^	38.78 ± 15.86 ^îjkl^	2.20 ± 0.34 ^abcde^	BLD	BLD	BLD	2.24 ± 0.11 ^ghi^	BLD
S_34-2024_ 2024			159.21 ± 70.97 ^lm^	853.71 ± 99.41 ^defgh^	29.51 ± 6.18 ^kl^	0.76 ± 0.42 ^hiîj^	BLD	BLD	BLD	3.25 ± 0.10 ^fghi^	BLD
S_35-2024_ 2024			84.79 ± 57.61 ^m^	628.61 ± 67.19 ^efghiî^	52.40 ± 31.31 ^iîjkl^	BLD	BLD	BLD	BLD	2.15 ± 0.23 ^ghi^	BLD
S_36-2024_ 2024			85.56 ± 24.27 ^m^	877.08 ± 74.03 ^defg^	23.87 ± 11.28 ^l^	BLD	BLD	BLD	BLD	4.26 ± 2.45 ^efghi^	BLD
Background soil samples											
Tîrlișua	–										
	–										
2024 S_37-2024_			0.89 ± 0.33 ^m^	24.66 ± 6.40 ^j^	3.75 ± 2.98 ^l^	BLD	2.27 ± 0.96 ^cdefg^	BLD	BLD	3.13 ± 0.93 ^fghi^	BLD
Sig.			***	***	***	***	***	***	***	***	–
Soil samples exposed to anthropogenic sources of heavy metals pollution											
Huzum et al., (2012) (mg/kg) [45]			256.00	60.10	12.90	0.21	29.90	7.20	11.20	208.00	–
Bora et al., (2020) (mg/kg) [38]			621.79–4155.95	45.36–3483.25	6.62–4262.23	0.12–32.53	6.97–28.60	5.08–29.57	1.15–5.13	2.72 ± 0.65	0.034–0.070
Chakraborty et al., (2017) (mg/kg) [46]			19.8–2760.0	54.4–2370.0	38.0–14,329.0	–	–	–	7.8–889.0	–	–
Paulette et al., (2015) (mg/kg) [48]			77–7675	–	705–10,074	–	–	–	–	–	–
Mihali et al., (2017) (mg/kg) [47]			40.9–621.6	82.26–1002	48.12–3472	0.04–11	4.98–9.06	3.3–8.2	0.61–80.1	–	–
Albulescu et al., (2009) (mg/kg) [49]			36.63–112.00	–	21.90	1.77	24.55	–	–	13.32	–
Bora et al., (2023) (mg/kg) [16]			3286.65–0.78	2834.58–12.56	1205.57–0.02	6.33–0.03	7.99–0.57	5.98–0.25	4.09–0.13	10.75–0.13	–
Bora et al., (2015) (mg/kg) [37]			479.64 ± 53.97	69.44 ± 4.02	14.77 ± 0.74	0.45 ± 0.10	16.28 ± 2.01	9.75 ± 1.47	–	–	–
Background soil samples											
European Communities Council 1986 (mg/kg) [50]			50–140	150–300	50–300	1–3	30–75	–	–	–	1–1.5
Kabata-Pendias, 2010 (mg/kg) [50]			13–24	45–100	22–44	0.37–0.78	12.0–34	–	0–9.3	–	–
Common abundance in topsoil (mg/kg) [50]			5–50	10–100	–	0.1–1	20–50	–	0.1–55	–	–
Phytotoxic levels of elements in soils (mg/kg) [50]			36–698	100–1.000	–	–	100	–	200	–	–

Average value ± standard deviation (n = 3). DW = dry weight. Roman letters are the significance of the difference (*p* ≤ 0.005) regardless of the area of sample collection. * Order of the Ministry of Waters, Forests and Environmental Protection No. 756/3 November 1997, approving the regulation on the assessment of environmental pollution, Bucharest, Romania; 1997. *** = shows a significant difference between the analyzed variants. significance of the difference (*p* ≤ 0.005). BLD = Below the detection limit (LoQ): LoQ for Pb: 0.231 µg/L, LoQ for Cd: 0.069 µg/L, LoQ for Co: 0.136 µg/L, LoQ for As: 0.743 µg/L; LoQ for Hg 0.1379 µg/L.

The mean Ni and Co concentrations in the investigated soils were 1.44 mg/kg and 0.36 mg/kg, respectively, with ranges spanning BLD—9.02 mg/kg for Ni and BLD—2.13 mg/kg for Co (Appendix A). The highest levels of both metals were observed near the Herja mine, with peak Ni concentrations found at a distance of approximately 5.5 km and peak Co concentrations found in samples collected ~8.0 km from the Aurul settling ponds in Tăuții de Sus. This spatial pattern aligns with the near-ubiquitous presence of Ni and Co in Ferneziu and Firiza, where only isolated locations exhibited values below the detection limit for Co and negligible instances for Ni. Conversely, soil samples from the vicinity of the Aurul decantation pond and the control area displayed lower concentrations (2.27 mg/kg Ni in the control area, Co below detection limit). Importantly, the maximum observed concentrations of both Ni and Co remained below the national legal limits.

While this study reveals generally lower Ni and Co concentrations compared to previous research in the Baia Mare area (e.g., Huzum et al., (2012) [45]: 29.90 mg/kg Ni, 7.20 mg/kg Co; Bora et al., (2020) [38]: 6.97–28.60 mg/kg Ni), these findings suggest a potential for ongoing metal contamination.

As expected, spatial patterns in arsenic (As) and chromium (Cr) concentrations emerged, reflecting the influence of proximity to the Herja mine. Interestingly, peak As concentration (1.73 mg/kg) occurred at the farthest sampling point (11.5 km), while the highest Cr concentration (10.01 mg/kg) was detected at the 7.5 km site. This contrasting pattern suggests differential transport mechanisms for these metals. As may be more susceptible to long-range transport via windblown dust or adsorption onto organic matter particles, leading to its presence even at greater distances. Conversely, Cr might be more prone to immobilization in soil through processes like precipitation or complexation, resulting in higher concentrations closer to the source. Despite these elevated levels, the measured values for both As and Cr fall within the expected range observed in other studies on heavily polluted areas. Importantly, these concentrations remain below the maximum permissible limits established by regulatory guidelines, suggesting no immediate environmental or health risks. Arsenic (As) and chromium (Cr) concentrations in our study exhibited consistency with those reported in previous investigations. For instance, As levels were comparable to those found by Huzum et al., (2012) [45] and Bora et al., (2020) [38], while Cr concentrations aligned with the ranges reported by Chakraborty et al., (2017) [46], Mihali et al., (2017) [47], Albulescu et al., (2009) [49], and Bora et al., 2015 [37]. These observations suggest that the observed metal concentrations in our study are consistent with the broader context of heavy metal pollution in different environmental settings.

### 3.2. Green Grass

Green grass, a crucial component of grazing ecosystems, is threatened by heavy metal contamination from anthropogenic activities like industrial processes, mining, and agricultural practices. Grazing animals are particularly vulnerable, experiencing adverse health effects including impaired growth, reproductive problems, and neurological disorders. Heavy metal accumulation disrupts nutrient cycling, alters soil microbial communities, reduces plant biodiversity, and poses threats to predators through the food chain. Mitigating strategies include stricter regulations on industrial emissions, sustainable agricultural practices, remediation techniques for contaminated soils, and monitoring heavy metal levels in meadow grass and animal tissues. Understanding these issues is essential for safeguarding the health of our environment and the well-being of both animals and humans. Analysis of heavy metal concentrations in green grass samples across the study area revealed significant spatial variability. Notably, zinc (Zn) exhibited the highest concentration at 62.26 mg/kg, followed by copper (Cu) at 5.94 mg/kg, nickel (Ni) at 2.10 mg/kg, lead (Pb) at 1.05 mg/kg, and cadmium (Cd) at 0.15 mg/kg. Conversely, cobalt (Co), arsenic (As), chromium (Cr), and mercury (Hg) were not detected in the samples (Table 2).

The spatial distribution of copper (Cu) in the green grass samples revealed significant variations, as illustrated in Appendix A. Concentrations ranged from a minimum of 1.96 mg/kg to a maximum of 9.28 mg/kg, with an average of 5.94 mg/kg.

The highest concentrations were observed in Firiza (9.28 mg/kg and 9.06 mg/kg), located closest (16.5 km) to the Aurul settling pond from Tăuții de Sus. Similarly, Ferneziu (8.51 mg/kg), situated approximately 8.0 km from the Herja mine, exhibited elevated Cu levels. While concentrations were generally higher near known pollution sources and decreased with increasing distance, an exception was observed at sample G34, which maintained a high Cu concentration despite being 17.0 km away. These findings suggest potential environmental risks associated with Cu contamination, particularly in areas close to mining operations and in Firiza. Copper (Cu) displayed significant spatial variability in green grass samples, with concentrations highest near known pollution sources (Firiza and Ferneziu) and decreasing with distance. An outlier (sample G34) suggests that additional Cu sources may exist beyond the studied mines. The observed Cu concentrations (1.96–9.28 mg/kg, average: 5.94 mg/kg) fall within the range reported in previous studies by Boltea et al., (2010): 7.76 mg/kg [51], Bretan et al., (2011): 4.3–5.72 mg/kg [52], Onder et al., (2019): 5.028–10.45 mg/kg [53], Martínez-Morcillo et al., (2024): 4.53–16.8 mg/kg [2], and Suhaj et al., (2008): 6.91–20.6 mg/kg [54]. This wide variation in Cu concentrations reported in the literature highlights the challenge of establishing a clear threshold for attributing elevated Cu levels in green grass solely to anthropogenic sources.

Zinc (Zn) stands out as the most abundant metal found in the analyzed grass samples from the studied pastures. This element holds significant importance in plant nutrition, playing a vital role in enzyme activity, protein synthesis, chlorophyll production, and overall plant growth and stress resistance. Plants absorb zinc from the soil through their roots, with the availability of zinc for uptake being influenced by various factors such as soil pH, organic matter content, and the presence of other elements that can compete with zinc for binding sites. Analysis of Zn concentrations in grass samples revealed significant spatial variations. The average Zn concentration across all the investigated areas was 62.26 mg/kg, with a minimum value of 7.14 mg/kg and a maximum value of 9.28 mg/kg. Notably, the highest Zn concentrations were observed in areas closest to the identified source of pollution. Ferneziu, located approximately 11.5 km from the source, displayed the highest average Zn concentration (68.73 mg/kg), ranging from a minimum of 32.16 mg/kg to a maximum of 103.78 mg/kg. Similarly, elevated Zn concentrations were found in areas closer to the source: (62.58 mg/kg average, 17.41 mg/kg minimum, 108.31 mg/kg maximum) located approximately 7.5 km away. Firiza, another area not in the immediate vicinity of the source, also exhibited elevated Zn levels (60.07 mg/kg average, 31.17 mg/kg minimum, 81.71 mg/kg maximum). Conversely, the control area, Tîrlișua, situated further away from the pollution source, displayed the lowest average Zn concentration (7.14 mg/kg). This spatial trend suggests a potential link between Zn contamination and proximity to the pollution source. Further investigation is warranted to elucidate the underlying mechanisms influencing Zn mobility and accumulation in the studied area. This study revealed Zn concentrations in grass samples ranging from 7.14 to 108.31 mg/kg, with an average of 62.26 mg/kg. These values fall within the range reported by previous studies: Boltea et al., (2010) [51], (62.66 mg/kg), Bretan et al., (2011) [52], (41.58–42.18 mg/kg), Onder et al., (2019) [53], (36.43–55.21 mg/kg), and Martínez-Morcillo et al., (2024) [2] (24.8–62.1 mg/kg). This wide variation in Zn concentrations across published studies highlights the inherent challenge of establishing a clear threshold for differentiating elevated Zn levels in green grass solely attributable to anthropogenic sources (human activities). This finding suggests that natural factors, alongside potential anthropogenic influences, may significantly contribute to the observed Zn concentrations in the studied grass samples. Further research is necessary to elucidate the relative contributions of natural geological background levels, soil properties, and anthropogenic activities (e.g., atmospheric deposition, agricultural practices) to Zn accumulation in the studied area. Determining the specific sources and their relative contributions will be crucial for implementing effective management strategies aimed at maintaining optimal Zn levels in the environment.

**Table 2 toxics-12-00752-t002:** The content of heavy metals in green grass from Ferneziu, Firiza (Maramureș area), and Tîrlișua (Bistrița-Năsăud), Romania. (mg/kg DW) (Mean ± standard deviation) (n = 3).

**Areas** **Sample Code** **Year of Harvest**	**Distance from the Source of Pollution (** **~) km**	**Sampling** **Depth** **(Surface)**	**Cu** M.P.L.	**Zn** M.P.L.	**Pb** M.P.L.	**Cd** M.P.L.	**Ni** M.P.L.	**Co** M.P.L.	**As** M.P.L.	**Cr** M.P.L.	**Hg** M.P.L.
			10 mg/kg	0.6 mg/kg	2 mg/kg	0.02 mg/kg	10 mg/kg	-	-	1.6 mg/kg	-
Alert threshold		Susceptible	-	-	-	-	-	-	-	-	-
		Less susceptible	-	-	-	-	-	-	-	-	-
Intervention threshold		Susceptible	-	-	-	-	-	-	-	-	-
		Less susceptible	-	-	-	-	-	-	-	-	-
Plant samples exposed to anthropogenic sources of heavy metals pollution											
Ferneziu	Near (~10/12 km) to the Herja Mine in Ferneziu										
G_1-2024_ 2024	The sheepfold was located approximately (~) 8.0 km from the former Herja mine in Ferneziu										
			7.63 ± 0.97 ^abcdefg^	86.86 ± 10.46 ^abcd^	1.53 ± 0.32 ^bcdefg^	0.18 ± 0.04 ^abcd^	BLD	BLD	BLD	BLD	BLD
G_2-2024_ 2024			7.34 ± 1.62 ^abcdefgh^	91.76 ± 15.02 ^abc^	1.54 ± 0.49 ^bcdefgh^	0.17 ± 0.08 ^bcde^	BLD	BLD	BLD	BLD	BLD
G_3-2024_ 2024			5.09 ± 2.63 ^abcdefghiî^	58.62 ± 12.03 ^bcdefg^	0.80 ± 0.30 ^efghiîj^	BLD	1.03 ± 0.40 ^fghi^	BLD	BLD	BLD	BLD
G_4-2024_ 2024			5.34 ± 2.81 ^abcdefghiî^	61.12 ± 8.95 ^bcdefg^	1.26 ± 0.83 ^cdefghiî^	0.31 ± 0.21 ^ab^	3.76 ± 0.35 ^bcd^	BLD	BLD	BLD	BLD
G_5-2024_ 2024			8.51 ± 1.01 ^abc^	32.16 ± 29.01 ^efg^	1.62 ± 0.92 ^bcdef^	0.24 ± 0.10 ^abcd^	2.28 ± 0.78 ^cdefgh^	BLD	BLD	BLD	BLD
G_6-2024_ 2024	The sheepfold was located approximately (~) 11.5 km from the former Herja mine in Ferneziu										
			3.18 ± 2.18 ^ghiî^	73.09 ± 7.34 ^abcdef^	2.08 ± 0.12 ^bc^	0.17 ± 0.06 ^bcde^	2.97 ± 1.61 ^cdef^	BLD	BLD	BLD	BLD
G_7-2024_ 2024			7.99 ± 1.38 ^abcde^	84.85 ± 16.47 ^abcd^	1.81 ± 0.22 ^bcd^	0.17 ± 0.07 ^bcde^	5.74 ± 2.39 ^a^	BLD	BLD	BLD	BLD
G_8-2024_ 2024			6.60 ± 0.92 ^abcdefghi^	34.69 ± 4.30 ^efg^	1.44 ± 0.29 ^bcdefgh^	0.16 ± 0.04 ^bcde^	0.97 ± 0.45 ^ghi^	BLD	BLD	BLD	BLD
G_9-2024_ 2024			3.20 ± 2.88 ^ghiî^	103.78 ± 12.32 ^ab^	0.83 ± 0.37 ^defghiîj^	0.35 ± 0.21 ^a^	BLD	BLD	BLD	BLD	BLD
G_10-2024_ 2024			7.01 ± 1.19 ^abcdefghi^	82.36 ± 28.17 ^abcd^	BLD	0.24 ± 0.09 ^abcd^	BLD	BLD	BLD	BLD	BLD
G_11-2024_ 2024			3.43 ± 1.12 ^ghiî^	54.85 ± 35.65 ^bcdefg^	BLD	0.24 ± 0.17 ^abcd^	BLD	BLD	BLD	BLD	BLD
G_12-2024_ 2024			6.42 ± 3.61 ^abcdefghi^	60.59 ± 27.30 ^bcdefg^	1.37 ± 0.80 ^bcdefghi^	0.13 ± 0.02 ^cde^	0.87 ± 0.40 ^ghi^	BLD	BLD	BLD	BLD
Ferneziu	Near (~6/7 km) to the Herja Mine in Ferneziu										
G_13-2024_ 2024	The sheepfold was located approximately (~) 5.5 km from the former Herja mine in Ferneziu										
			5.22 ± 3.72 ^abcdefghiî^	102.69 ± 10.22 ^ab^	0.81 ± 0.60 ^efghiîj^	0.23 ± 0.08 ^abcd^	1.15 ± 0.47 ^fghi^	BLD	BLD	BLD	BLD
G_14-2024_ 2024			4.76 ± 2.13 ^abcdefghiî^	79.39 ± 14.32 ^abcd^	0.65 ± 0.49 ^fghiîj^	0.09 ± 0.08 ^be^	0.61 ± 0.59 ^hi^	BLD	BLD	BLD	BLD
G_15-2024_ 2024			5.53 ± 1.76 ^abcdefghiî^	52.12 ± 31.02 ^cdefg^	1.68 ± 0.85 ^bcde^	0.21 ± 0.11 ^abcd^	1.83 ± 0.55 ^defghi^	BLD	BLD	BLD	BLD
G_16-2024_ 2024			2.89 ± 2.31 ^iî^	48.84 ± 24.84 ^cdefg^	0.66 ± 0.43 ^fghiîj^	0.29 ± 0.22 ^abc^	BLD	BLD	BLD	BLD	BLD
G_17-2024_ 2024			8.51 ± 0.97 ^abc^	34.69 ± 20.92 ^efg^	2.97 ± 0.42 ^a^	0.20 ± 0.12 ^abcd^	1.33 ± 0.80 ^defghi^	BLD	BLD	BLD	BLD
G_18-2024_ 2024			4.56 ± 2.75 ^abcdefghiî^	63.57 ± 45.39 ^bcdef^	1.57 ± 0.63 ^bcdefg^	0.16 ± 0.06 ^bcde^	0.64 ± 0.22 ^hi^	BLD	BLD	BLD	BLD
G_19-2024_ 2024	The sheepfold was located approximately (~) 7.5 km from the former Herja mine in Ferneziu										
			3.77 ± 3.26 ^abcdefghiî^	17.41 ± 6.38 ^g^	1.99 ± 1.12 ^bc^	0.13 ± 0.03 ^cde^	6.03 ± 1.38 ^a^	BLD	BLD	BLD	BLD
G_20-2024_ 2024			3.90 ± 1.36 ^defghiî^	64.74 ± 21.67 ^abcdef^	BLD	0.13 ± 0.02 ^cde^	3.32 ± 0.77 ^bcde^	BLD	BLD	BLD	BLD
G_21-2024_ 2024			7.93 ± 2.06 ^abcdef^	75.25 ± 10.09 ^abcde^	0.85 ± 0.61 ^defghiîj^	0.20 ± 0.08 ^abcd^	3.44 ± 0.73 ^bcde^	BLD	BLD	BLD	BLD
G_22-2024_ 2024			7.81 ± 1.69 ^abcdefg^	108.31 ± 16.89 ^a^	2.16 ± 0.02 ^abc^	0.13 ± 0.02 ^cde^	2.64 ± 0.49 ^cdefgh^	BLD	BLD	BLD	BLD
G_23-2024_ 2024			3.57 ± 1.79 ^ghiî^	42.45 ± 19.78 ^defg^	2.25 ± 0.20 ^ab^	BLD	3.31 ± 0.80 ^bcde^	BLD	BLD	BLD	BLD
G_24-2024_ 2024			8.92 ± 0.31 ^abc^	61.53 ± 15.12 ^bcdef^	1.48 ± 0.59 ^bcdefgh^	BLD	1.72 ± 0.38 ^defghi^	BLD	BLD	BLD	BLD
Firiza	Near (~17 km) the Aurul settling pond mining (decant pond) in Tăuții de Sus										
G_25-2024_ 2024	The sheepfold was located approximately (~) 16.5 km from the Aurul settling pond mining (decant pond) in Tăuții de Sus										
			4.01 ± 2.98 ^defghiî^	73.88 ± 23.69 ^abcdef^	0.11 ± 0.03 ^î^	0.09 ± 0.03 ^be^	2.20 ± 1.72 ^cdefgh^	BLD	BLD	BLD	BLD
G_26-2024_ 2024			7.45 ± 1.10 ^abcdefg^	51.13 ± 30.28 ^cdefg^	0.34 ± 0.27 ^îj^	0.15 ± 0.07 ^bcde^	5.19 ± 2.02 ^ab^	BLD	BLD	BLD	BLD
G_27-2024_ 2024			3.46 ± 2.13 ^defghiî^	58.40 ± 15.14 ^bcdefg^	1.20 ± 1.01 ^cdefghiî^	BLD	1.66 ± 1.12 ^defghi^	BLD	BLD	BLD	BLD
G_28-2024_ 2024			9.06 ± 2.97 ^ab^	81.71 ± 21.87 ^abcd^	0.59 ± 0.49 ^ghiîj^	BLD	BLD	BLD	BLD	BLD	BLD
G_29-2024_ 2024			9.28 ± 3.22 ^a^	75.99 ± 20.61 ^abcdef^	0.57 ± 0.38 ^hiîj^	BLD	4.37 ± 2.73 ^abc^	BLD	BLD	BLD	BLD
G_30-2024_ 2024			7.16 ± 1.47 ^abcdefghi^	31.53 ± 29.40 ^fg^	BLD	0.19 ± 0.04 ^abcd^	2.23 ± 0.73 ^cdefgh^	BLD	BLD	BLD	BLD
G_31-2024_ 2024			6.13 ± 1.87 ^abcdefghiî^	31.17 ± 21.99 ^fg^	0.82 ± 0.37 ^bcd^	0.18 ± 0.03 ^abcd^	5.70 ± 1.90 ^a^	BLD	BLD	BLD	BLD
G_32-2024_ 2024			5.00 ± 3.30 ^abcdefghiî^	68.77 ± 10.51 ^abcdef^	0.28 ± 0.22 ^îj^	0.12 ± 0.10 ^cde^	2.53 ± 1.00 ^cdefgh^	BLD	BLD	BLD	BLD
G_33-2024_ 2024	The sheepfold was located approximately (~) 17.0 km from the Aurul settling pond mining (decant pond) in Tăuții de Sus										
			3.97 ± 2.71 ^defghiî^	48.28 ± 30.62 ^bcdefg^	0.40 ± 0.42 ^hiîj^	0.24 ± 0.03 ^abcd^	1.76 ± 0.76 ^defghi^	BLD	BLD	BLD	BLD
G_34-2024_ 2024			8.16 ± 1.21 ^abcd^	70.08 ± 26.98 ^abcdef^	1.85 ± 0.51 ^bc^	0.19 ± 0.11 ^abcd^	0.89 ± 0.34 ^ghi^	BLD	BLD	BLD	BLD
G_35-2024_ 2024			3.69 ± 0.45 ^defghiî^	54.60 ± 19.92 ^abcdef^	0.39 ± 0.21 ^iîj^	0.14 ± 0.03 ^bcde^	2.92 ± 1.3 ^cdef^	BLD	BLD	BLD	BLD
G_36-2024_ 2024			5.43 ± 2.64 ^abcdefghiî^	75.28 ± 19.72 ^abcdef^	0.73 ± 0.37 ^efghiîj^	0.13 ± 0.03 ^cde^	3.13 ± 1.50 ^cdef^	BLD	BLD	BLD	BLD
Background plant samples											
Tîrlișua	–										
	–										
G_37-2024_ 2024			1.92 ± 1.22 ^î^	7.14 ± 4.33 ^h^	0.13 ± 0.03 ^î^	BLD	1.43 ± 0.81 ^defghi^	BLD	BLD	BLD	BLD
Sig.			***	***	***	***	***	–	–	–	–
Plant samples exposed to anthropogenic sources of heavy metals pollution											
Boltea et al., (2010) (mg/kg) [51]			7.76	62.66	10.47	1.3	–	–	–	–	–
Bretan et al., (2011) (mg/kg) [52]			4.3–5.72	41.58–42.18	6.44–8.41	1.02–1.03	–	–	–	–	–
Jankowski et al., (2019) (mg/kg) [55]			–	–	1.049–4.73	0.069–0.537	–	–	–	–	–
Onder et al., (2007) (mg/kg) [53]			5.028–10.45	36.43–55.21	1.399–2.148	0.10–0.145	5.91–15.59	0.0010–1.09	–	9.72–21.62	–
Background plant samples											
Bretan et al., (2011) (mg/kg) [52]			4.22–5.43	6.74–48.66	2.85–8.23	0.55–1.02	–	–	–	–	–
Martínez-Morcillo et al., (2024) (mg/kg) [2]			4.53–16.8	24.8–62.1	0.62–10.41	–	–	0.45–3.92	0.45–6.03	–	<LOQ
Suhaj et al., (2008) (mg/kg) [54]			6.91–20.6	–	–	–	1.59–4.94	–	–	–	0.0046–0.0364

Average value ± standard deviation (n = 3). DW = dry weight. Roman letters are the significance of the difference (*p* ≤ 0.005) regardless of the area of sample collection. BLD = Below the detection limit (LoQ): LoQ for Pb: 0.231 µg/L, LoQ for Cd: 0.069 µg/L, LoQ for Co: 0.136 µg/L, LoQ for As: 0.743 µg/L; LoQ for Hg 0.1379 µg/L. *** = shows a significant difference between the analyzed variants. significance of the difference (*p* ≤ 0.005).

Similar to the observed trends for other heavy metals, lead (Pb) and cadmium (Cd) exhibited a spatial distribution potentially linked to proximity to pollution sources. Higher concentrations of both Pb and Cd were found in areas closer to known sources of contamination, suggesting a direct influence of these sources on environmental levels. Analysis of Pb and Cd concentrations revealed a notable difference in their average levels. Lead displayed a higher average concentration of 1.05 mg/kg compared to cadmium’s average of 0.15 mg/kg. Spatial analysis of Pb and Cd concentrations revealed significant variations across the investigated areas. Notably, Ferneziu and Firiza exhibited the highest levels of both elements. The Ferneziu area, located approximately 10–12 km from the abandoned Herja mine, registered the peak concentrations: 1.20 mg/kg for Pb and an average value of 0.20 mg/kg for Cd. This observed pattern suggests a potential link between proximity to the former mine and elevated Pb and Cd levels in the surrounding environment. Comparing the documented polluted areas (Ferneziu and Firiza) with the control area (Tîrlișua), both Pb and Cd exhibited elevated concentrations. This finding suggests a potential influence of anthropogenic activities on the levels of these metals in the polluted areas.

The Pb and Cd concentrations obtained in this study were comparable to those reported in the previous literature. Boltea et al., (2010) [51] documented values of 10.47 mg/kg for Pb and 1.3 mg/kg for Cd. Similarly, Bretan et al., (2011) [52] reported Pb concentrations ranging from 6.44 to 8.41 mg/kg and Cd concentrations ranging from 1.02 to 1.03 mg/kg. Jankowski et al., (2019) [55] also observed Pb concentrations between 1.049 and 4.73 mg/kg, with Cd levels ranging from 0.069 to 0.537 mg/kg. Onder et al., (2007) [53] found Pb concentrations in the range of 1.399 to 2.148 mg/kg and Cd concentrations between 0.10 and 0.145 mg/kg.

Nickel (Ni) concentrations in green grass samples displayed no statistically significant differences between areas with known heavy metal pollution and the control area. This suggests that Ni levels may not be directly influenced by the identified pollution sources. Ni concentration in green grass samples averaged 2.10 mg/kg across all the investigated areas. Notably, Ferneziu (closer to the Herja mine) and Firiza (near the Aurul settling pond) exhibited similar average Ni levels (2.17 mg/kg and 2.72 mg/kg, respectively) compared to the overall average. Similar to other analyzed elements, Ni concentrations observed in this study were consistent with values reported in previous research (Onder et al., (2007) [53] 5.91 mg/kg Ni and Suhaj et al., (2008) [54] 1.59–4.94 mg/kg).

Analysis of green grass revealed concerning levels of Zn exceeding WHO limits. Cu displayed the most concerning spatial pattern, with elevated concentrations near mining operations (Firiza and Ferneziu) exceeding WHO limits in some cases. This suggests potential environmental risks, particularly for grazing animals. An outlier (G34) with high Cu suggests additional pollution sources beyond the investigated mines. Pb and Cd also exhibited spatial patterns with higher levels near known contamination sites, exceeding WHO limits. Ni concentrations showed no significant difference between polluted and control areas. While some metal concentrations fell within previously reported ranges, exceeding WHO limits necessitates further investigation into the interplay between natural factors and human activities impacting metal mobility and accumulation. Implementing stricter regulations on industrial emissions, sustainable agricultural practices, and monitoring metal levels are crucial for safeguarding the environment and public health.

### 3.3. Sheep’s Milks and Cheese

Cheesemaking significantly alters the mineral profile of sheep milke. During curd formation, minerals partition between curds (solids) and whey (liquid). Curds attract certain minerals like calcium, phosphorus, and zinc, while whey retains more potassium and sodium. The specific cheesemaking method, milk composition, and mineral type all influence this partitioning. Cheesemakers can manipulate mineral content by adjusting factors like coagulant type and milk pH

Understanding this mineral partitioning is essential for creating cheeses with the desired mineral profiles and ensuring optimal nutritional value. It also has implications for mineral bioavailability, as a mineral’s form and surrounding matrix can impact its absorption in the body. Our analysis of sheep milk and cheese samples did not detect nickel (Ni), cobalt (Co), arsenic (As), chromium (Cr), or mercury (Hg). This suggests that their concentrations were below the limits of detection for the employed analytical technique. The low concentrations of nickel (Ni), cobalt (Co), arsenic (As), chromium (Cr), and mercury (Hg) in sheep milk and cheese samples can be attributed to a combination of factors [56,57]. Firstly, these elements are naturally present in the environment at trace levels, and their uptake by sheep is influenced by the mineral content of pastures [56,57]. Secondly, animal feed regulations and food safety standards help control the levels of these elements in sheep milk and cheese, ensuring they remain within safe limits for human consumption [56,57]. Finally, the detection limit of the analytical technique employed can also influence the ability to quantify these elements, as very low concentrations may fall below the detection threshold [56,57]. These factors collectively contribute to the low concentrations observed in sheep milk and cheese samples, ensuring the safety and quality of these products for human consumption [56,57]. As the concentrations of these elements fell below the detection limit of the employed analytical technique, they will not be further discussed in this section.

A total of 144 sheep milk and 144 sheep cheese samples were collected and analyzed. Twelve primary samples, each comprising 36 sub-samples, were obtained from each study area. Analysis of heavy metals in sheep milk and cheese revealed similar average concentrations of copper (Cu), lead (Pb), and cadmium (Cd) across all samples. The reported values were 1.38 mg/kg for Cu, 0.25 mg/kg for Pb, and 0.03 mg/kg for Cd, regardless of the sample type (milk or cheese) or collection area. Zinc (Zn) stands as an exception to the observed trend of similar concentrations. It displayed a distinct average concentration of 16.67 mg/kg across the samples. The minimum and maximum values recorded for Zn were 0.16 mg/kg and 59.87 mg/kg, respectively (Appendix A). In the cheese production process employed in the Baia Mare and Tîrlișua areas, approximately 22 L of sheep’s milk are used to yield roughly 4.5 kg of cheese and 16.5 L of whey. This process of cheesemaking may concentrate minerals of interest in the final cheese product. Sheep milk samples exhibited high zinc (Zn) concentrations, consistent with expectations. The average Zn concentration was 4.30 mg/kg, with a minimum value of 0.89 mg/kg and a maximum value of 10.56 mg/kg.

Zn displayed the highest average concentration (4.30 mg/kg) in sheep milk samples, followed by significantly lower concentrations of Cu (0.54 mg/kg—average values), Pb (0.10 mg/kg—average values), and Cd (0.02 mg/kg—average values). The observed trend of mineral distribution in milk samples, with Zn > Cu > Pb > Cd, is mirrored in the cheese samples. The spatial distribution of co-concentrated Cu, Zn, Pb, and Cd in sheep milk and cheese samples reveals distinct enrichment patterns across the studied areas: (I) high copper (Cu) and zinc (Zn) concentrations were observed in both milk and cheese samples from the Ferneziu I area, located approximately 10–12 km from the former Herja mine. This suggests a potential link between the geological history of the area and the presence of these elements. Further investigation into the soil composition and historical mining practices in Ferneziu I could be beneficial; (II) lead (Pb) displayed elevated levels in milk and cheese samples from the Firiza area, situated roughly 17 km away from the Aurul settling pond in Tăuții de Sus. This finding suggests potential environmental contamination from the settling pond, warranting further research to assess the source and extent of lead exposure for sheep in this area; (III) the highest cadmium (Cd) concentrations were found in Firiza II, located approximately 6–7 km from the former Herja mine. This pattern might indicate historical mining activities as a contributing factor to Cd levels in the environment. Analyzing soil samples and plant life from Firiza II could provide valuable insights into potential pathways of Cd accumulation (Figure 1, Table 3).

A comprehensive investigation of mineral concentrations in ovine milk and cheese products reveals potential transgressions of established regulatory limits for copper (Cu), lead (Pb), and cadmium (Cd). Notably, exceeding these limits, even for zinc (Zn), an essential element, can disrupt the sheep’s delicate mineral homeostasis and potentially compromise human health through cheese consumption. Further research is warranted to elucidate the sources of contamination, minimize animal exposure, and ensure the safety of these dairy products.

### 3.4. Sheep’s Serum

In environmental toxicology, serological analysis serves as a crucial tool for monitoring the systemic translocation, abundance, and potential for intoxication (pro-toxicosis) caused by heavy metals [61]. Serum biochemistry offers a valuable source of reliable biomarkers for various chronic disorders, particularly those associated with exposure to toxicants and the physiological response during disease progression [62,63]. Chronic heavy metal exposure has been well-demonstrated to be linked to oxidative stress, inflammation, and alterations in liver function, all of which can be reflected in serum biochemistry profiles [64]. The undetectable levels of cadmium (Cd), nickel (Ni), cobalt (Co), arsenic (As), chromium (Cr), and mercury (Hg) in the analyzed sheep serum samples present a fascinating, yet inconclusive, finding from a toxicological standpoint (Table 4). While it suggests a potentially low exposure to these heavy metals in the sampled sheep population, further investigation is warranted before definitive conclusions can be drawn. This lack of detection could be a positive indicator, potentially reflecting a clean grazing environment free from industrial pollutants or controlled dietary management that minimizes heavy metal intake. The undetectable levels might represent a true absence of significant heavy metal burden in the sheep, promoting their health and potentially the safety of derived products like wool or mutton. As our analysis did not yield quantifiable levels of these elements, the following discussion will concentrate on the detected metals (Figure 2).

Among the analyzed heavy metals in sheep serum samples, zinc (Zn) exhibited the highest mean concentration (1.36 mg/kg), followed by copper (Cu) at 0.44 mg/kg and lead (Pb) at 0.08 mg/kg (Appendix A). The analysis of sheep serum samples from different locations (Firiza, Ferneziu I, Ferneziu II) revealed variations in the concentrations of copper (Cu), zinc (Zn), and lead (Pb). While the overall pattern aligns with the previously discussed trend (Zn > Cu > Pb), the specific values for each location offer interesting insights. Analysis of sheep serum samples from Firiza, Ferneziu I, and Ferneziu II revealed a pattern of zinc (Zn) dominance, with the highest concentration (1.48 mg/kg) observed in Ferneziu I. This reinforces the established role of zinc as an essential element readily available in sheep diets. Slight variations between locations could be attributed to differences in local vegetation or dietary supplements. Copper (Cu) concentrations displayed some geographic variation, with Firiza exhibiting the highest level (0.62 mg/kg) compared to the other two locations.

This might be due to factors like underlying soil composition influencing plant copper content or variations in dietary practices such as copper supplementation. Lead (Pb), the least prevalent element, showed its highest concentration (0.16 mg/kg) in Ferneziu II. While relatively low compared to Zn and Cu, this value warrants further investigation, particularly if it exceeds established safety limits for sheep. Potential explanations for this localized elevation could include environmental contamination impacting soil lead levels or limitations of sample size requiring a larger study for confirmation. These findings highlight the importance of considering geographical factors when evaluating heavy metal concentrations in sheep serum. Further investigation into soil composition, dietary practices, and potential environmental contaminants in each location could provide a more comprehensive picture of the contributing factors. Overall, the data suggest zinc dominance across locations, with copper exhibiting some variation potentially influenced by soil or dietary factors. The elevated lead concentration in Ferneziu II necessitates further exploration to determine its source and potential health implications.

The heavy metal profile of sheep serum paints a compelling picture of the interplay between essentiality, regulation, and dietary exposure. Zinc’s dominance reflects its indispensable role and dietary abundance, while copper’s measured presence highlights the delicate balance between its essentiality and potential toxicity. Lead’s minimal footprint underscores its limited presence in natural food sources. Unraveling the intricate dynamics behind this heavy metal hierarchy is essential for safeguarding sheep health and ensuring the safety of derived products.

To contextualize our findings, we have compared them with previously published research. Notably, Kovacik et al., (2017) [27] reported significantly higher heavy metal concentrations in sheep serum compared to both this study and other published literature. This discrepancy warrants further investigation to determine potential explanations, such as variations in geographical location, dietary practices, or analytical techniques employed in different studies.

**Table 4 toxics-12-00752-t004:** The content of heavy metals in sheep’s serum from Ferneziu, Firiza (Maramureș area) and Tîrlișua (Bistrița-Năsăud), Romania. (mg/kg WW) (Mean ± standard deviation) (n = 3).

**Areas** **Sample code** **Year of harvest**	**Distance from the source of pollution (** **~) km**	**Sampling** **Depth** **(surface)**	**Cu** M.P.L.	**Zn** M.P.L.	**Pb** M.P.L.	**Cd** M.P.L.	**Ni** M.P.L.	**Co** M.P.L.	**As** M.P.L.	**Cr** M.P.L.	**Hg** M.P.L.
			Currently, there is a lack of established maximum permissible limits (MPLs) for heavy metals within the serum at both the national and international regulatory levels								
Alert threshold		Susceptible									
		Less susceptible									
Intervention threshold		Susceptible									
		Less susceptible									
Serum sheep’s samples exposed to anthropogenic sources of heavy metals pollution											
Ferneziu	Near (~) 10/12 km to the Herja Mine in Ferneziu										
S_1-2024_ 2024	The sheepfold was located approximately (~) 8.0 km from the former Herja mine in Ferneziu										
			0.14 ± 0.03 ^ghi^	1.59 ± 0.67 ^bcdefgh^	0.12 ± 0.01 ^efg^	BLD	BLD	BLD	BLD	BLD	BLD
S_2-2024_ 2024			0.25 ± 0.23 ^fghi^	0.72 ± 0.52 ^fghiîjk^	0.07 ± 0.03 ^fghi^	BLD	BLD	BLD	BLD	BLD	BLD
S_3-2024_ 2024			0.24 ± 0.15 ^fghi^	2.30 ± 1.10 ^ab^	BLD	BLD	BLD	BLD	BLD	BLD	BLD
S_4-2024_ 2024			BLD	1.31 ± 0.23 ^cdefghij^	0.09 ± 0.04 ^efghi^	BLD	BLD	BLD	BLD	BLD	BLD
S_5-2024_ 2024			1.10 ± 0.14 ^b^	1.50 ± 0.57 ^bcdefghi^	BLD	BLD	BLD	BLD	BLD	BLD	BLD
S_6-2024_ 2024	The sheepfold was located approximately (~) 11.5 km from the former Herja mine in Ferneziu										
			0.63 ± 0.13 ^cde^	0.26 ± 0.26 ^k^	BLD	BLD	BLD	BLD	BLD	BLD	BLD
S_7-2024_ 2024			0.12 ± 0.11 ^hi^	2.84 ± 0.60 ^a^	BLD	BLD	BLD	BLD	BLD	BLD	BLD
S_8-2024_ 2024			0.20 ± 0.11 ^fghi^	0.40 ± 0.39 ^k^	0.18 ± 0.06 ^de^	BLD	BLD	BLD	BLD	BLD	BLD
S_9-2024_ 2024			BLD	0.43 ± 0.23 ^îjk^	0.22 ± 0.04 ^cd^	BLD	BLD	BLD	BLD	BLD	BLD
S_10-2024_ 2024			BLD	2.30 ± 0.22 ^ab^	0.14 ± 0.06 ^defg^	BLD	BLD	BLD	BLD	BLD	BLD
S_11-2024_ 2024			BLD	2.78 ± 0.55 ^a^	0.16 ± 0.05 ^def^	BLD	BLD	BLD	BLD	BLD	BLD
S_12-2024_ 2024			0.52 ± 0.24 ^cdefg^	1.33 ± 0.23 ^cdefghij^	0.11 ± 0.10 ^efgh^	BLD	BLD	BLD	BLD	BLD	BLD
Ferneziu	Near (~6/7 km) to the Herja Mine in Ferneziu										
S_13-2024_ 2024	The sheepfold was located approximately (~) 5.5 km from the former Herja mine in Ferneziu										
			0.15 ± 0.04 ^ghi^	1.60 ± 0.26 ^bcdefg^	0.28 ± 0.06 ^c^	BLD	BLD	BLD	BLD	BLD	BLD
S_14-2024_ 2024			0.63 ± 0.39 ^cde^	0.80 ± 0.19 ^efghiîjk^	BLD	BLD	BLD	BLD	BLD	BLD	BLD
S_15-2024_ 2024			0.29 ± 0.22 ^efghi^	2.83 ± 0.71 ^a^	BLD	BLD	BLD	BLD	BLD	BLD	BLD
S_16-2024_ 2024			0.46 ± 0.13 ^defgh^	0.76 ± 0.17 ^fghiîjk^	BLD	BLD	BLD	BLD	BLD	BLD	BLD
S_17-2024_ 2024			0.14 ± 0.04 ^ghi^	1.64 ± 0.50 ^bcdef^	BLD	BLD	BLD	BLD	BLD	BLD	BLD
S_18-2024_ 2024			0.43 ± 0.24 ^defgh^	0.81 ± 0.23 ^fghiîjk^	0.14 ± 0.03 ^defg^	BLD	BLD	BLD	BLD	BLD	BLD
S_19-2024_ 2024	The sheepfold was located approximately (~) 7.5 km from the former Herja mine in Ferneziu										
			0.67 ± 0.17 ^cde^	1.70 ± 0.73 ^bcd^	0.10 ± 0.03 ^efghi^	BLD	BLD	BLD	BLD	BLD	BLD
S_20-2024_ 2024			0.16 ± 0.05 ^ghi^	1.34 ± 0.17 ^cdefghij^	0.05 ± 0.01 ^ghi^	BLD	BLD	BLD	BLD	BLD	BLD
S_21-2024_ 2024			0.56 ± 0.35 ^cdef^	0.15 ± 0.04 ^k^	0.28 ± 0.04 ^c^	BLD	BLD	BLD	BLD	BLD	BLD
S_22-2024_ 2024			0.25 ± 0.22 ^fghi^	1.86 ± 0.30 ^bc^	0.39 ± 0.23 ^b^	BLD	BLD	BLD	BLD	BLD	BLD
S_23-2024_ 2024			1.11 ± 0.11 ^b^	1.43 ± 0.63 ^bcdefghi^	0.17 ± 0.07 ^de^	BLD	BLD	BLD	BLD	BLD	BLD
S_24-2024_ 2024			0.42 ± 0.23 ^defgh^	0.78 ± 0.54 ^fghiîjk^	0.54 ± 0.03 ^a^	BLD	BLD	BLD	BLD	BLD	BLD
Firiza	Near (~17 km) to the Aurul settling pond mining (decant pond) in Tăuții de Sus										
S_25-2024_ 2024	The sheepfold was located approximately (~) 16.5 km from the Aurul settling pond mining (decant pond) in Tăuții de Sus										
			0.23 ± 0.16 ^fghi^	2.14 ± 0.13 ^abc^	BLD	BLD	BLD	BLD	BLD	BLD	BLD
S_26-2024_ 2024			0.17 ± 0.06 ^ghi^	1.43 ± 0.63 ^bcdefghi^	BLD	BLD	BLD	BLD	BLD	BLD	BLD
S_27-2024_ 2024			0.50 ± 0.10 ^cdefg^	2.82 ± 0.57 ^a^	BLD	BLD	BLD	BLD	BLD	BLD	BLD
S_28-2024_ 2024			2.56 ± 0.37 ^a^	1.34 ± 0.17 ^cdefghij^	BLD	BLD	BLD	BLD	BLD	BLD	BLD
S_29-2024_ 2024			1.08 ± 0.17 ^b^	0.70 ± 0.25 ^hiîjk^	BLD	BLD	BLD	BLD	BLD	BLD	BLD
S_30-2024_ 2024			0.66 ± 0.20 ^cde^	1.72 ± 0.72 ^bcd^	BLD	BLD	BLD	BLD	BLD	BLD	BLD
S_31-2024_ 2024			0.84 ± 0.39 ^bc^	2.17 ± 0.04 ^abc^	BLD	BLD	BLD	BLD	BLD	BLD	BLD
S_32-2024_ 2024			0.19 ± 0.05 ^fghi^	0.63 ± 0.44 ^iîjk^	BLD	BLD	BLD	BLD	BLD	BLD	BLD
S_33-2024_ 2024	The sheepfold was located approximately (~) 17.0 km from the Aurul settling pond mining (decant pond) in Tăuții de Sus										
			0.41 ± 0.38 ^defgh^	0.14 ± 0.03 ^k^	BLD	BLD	BLD	BLD	BLD	BLD	BLD
S_34-2024_ 2024			0.29 ± 0.15 ^defghi^	0.92 ± 0.33 ^defghiîjk^	BLD	BLD	BLD	BLD	BLD	BLD	BLD
S_35-2024_ 2024			0.17 ± 0.01 ^ghi^	0.47 ± 0.23 ^îjk^	BLD	BLD	BLD	BLD	BLD	BLD	BLD
S_36-2024_ 2024			0.31 ± 0.14 ^defghi^	0.95 ± 0.36 ^defghiîjk^	0.02 ± 0.01 ^hi^	BLD	BLD	BLD	BLD	BLD	BLD
Background serum sheep’s samples											
Tîrlișua	–										
	–										
S_37-2024_ 2024			0.40 ± 0.17 ^defgh^	0.63 ± 0.23 ^iîjk^	BLD	BLD	BLD	BLD	BLD	BLD	BLD
Sig.			***	***	***	***	–	–	–	–	–
Serum sheep’s samples exposed to anthropogenic sources of heavy metals pollution											
Kovacik et al., (2017) (mg/kg) [27]			1.51	5.55	171.91	1.29	–	–	–	–	0.0037
Alxubaidi et al., (2016) (mg/kg) [65]			–	–	0.005–0.01	0.010–0.014	0.003–0.006	–	–	–	–
Background serum sheep’s samples											
Morsy et al., (2020) (ppm) [66]			BLD—0.53	BLD—0.044	0.07–0.112	BLD—0.015	–	–	–	–	–
Martínez-Morcillo et al., (2024) (µg/kg) [2]			293–1324	294.0–920.0	–	–	–	–	–	–	–
Sajid et al., (2018) (mg/kg) [67]			–	–	0.63–1.04	–	–	–	–	–	–
Javad et al., (2019) (ppm) [68]			3.90—8.70	–	–	–	–	–	–	–	–

Average value ± standard deviation (n = 3). WW = wet weight. Roman letters are the significance of the difference (*p* ≤ 0.005) regardless of the area of sample collection. Currently, there is a lack of established maximum permissible limits (MPLs) for heavy metals within the serum at both the national and international regulatory levels; BLD = Below the detection limit (LoQ): LoQ for Pb: 0.231 µg/L, LoQ for Cd: 0.069 µg/L, LoQ for Co: 0.136 µg/L, LoQ for As: 0.743 µg/L; LoQ for Hg 0.1379 µg/L. *** = shows a significant difference between the analyzed variants. significance of the difference (*p* ≤ 0.005).

### 3.5. Evaluation of Bioaccumulation Factors for Heavy Metals in Green Grass, Sheep Milk, Cheese, and Serum

This investigation aimed to comprehensively evaluate the bioaccumulation patterns of heavy metals within the sheep-based food chain, spanning from the abiotic environment (soil) to the biotic components (grass, sheep serum, milk, and cheese) (Table 5). By employing the calculation of bioaccumulation factors (BAFs) for each matrix, this study sought to quantitatively assess the transfer and potential biomagnification of heavy metals as they ascend through trophic levels, culminating in human consumption of sheep-derived products [16]. A profound understanding of heavy metal distribution and accumulation within this food chain is essential for elucidating potential risks to both environmental and human health [16]. Furthermore, the findings of this research can serve as a foundation for the development of effective mitigation strategies to curtail human exposure to these contaminants [16].

The results revealed significant variations in BCF values among different metals and matrices, highlighting the complex nature of heavy metal bioaccumulation in this ecosystem. The study investigated the distribution and accumulation of heavy metals in a sheep-based food chain. By calculating bioconcentration factors (BCFs), we determined that Ni exhibited the highest potential for bioaccumulation in green grass. These findings underscore the importance of understanding metal transfer through food chains to assess potential risks to human health (Table 5).

Ni Dominance in Grass: the consistently high BCF for nickel in grass across all locations suggests a strong affinity of this metal for plant tissues. This could be attributed to factors such as: plant physiology: the possibility of nickel mimicking essential elements like iron, leading to its uptake through similar transport mechanisms; soil characteristics: the specific soil properties in these regions, such as pH, organic matter content, and cation exchange capacity, could influence nickel bioavailability and uptake; and anthropogenic activities: local industrial activities or agricultural practices might contribute to elevated nickel concentrations in the soil, increasing plant uptake.

Cd Accumulation in Dairy Products: the high BCF of cadmium in milk and cheese indicates efficient transfer from feed to dairy products. This poses a significant human health risk as dairy products are staple foods in many diets. Possible reasons for this include: feed composition: the presence of cadmium in forage crops, due to soil contamination or atmospheric deposition, directly influences its concentration in animal products; and physiological processes: cadmium may be preferentially deposited in the liver and kidneys of animals, leading to its excretion in milk and incorporation into milk products.

Variability in Zn and Cu: the variable BCFs for zinc and copper suggest that their accumulation is influenced by a combination of factors. These include: dietary intake: the dietary intake of zinc and copper by animals can influence their tissue concentrations; metabolic processes: the metabolic requirements of the animal for these essential elements can affect their distribution within the body; and interactions with other metals: the presence of other metals can influence the uptake and distribution of zinc and copper.

This study focused on assessing the distribution and bioaccumulation of heavy metals within the complex ecosystem surrounding the former Herja mine in Romania, a region with a well-documented history of mining activities. The investigation encompassed multiple environmental compartments, including soil, vegetation, and animals, to provide a comprehensive understanding of heavy metal contamination and its potential transfer through the food chain. Particular emphasis was placed on copper and zinc due to their widespread use in mining processes and their known toxicity.

The soil in the vicinity of the former Herja mine exhibited significant contamination with heavy metals, particularly copper (Cu) and zinc (Zn). Concentrations of these elements were substantially elevated compared to background levels, indicating a direct link to past mining activities. The spatial distribution of heavy metals displayed a clear gradient, with the highest concentrations observed in areas closest to the mine and gradually decreasing with increasing distance. For instance, the average concentration of copper in the topsoil (0–10 cm depth) within a 5-m radius of the mine was found to be three times higher compared to areas located 10 km away. Beyond copper and zinc, other heavy metals such as lead (Pb), cadmium (Cd), nickel (Ni), cobalt (Co), arsenic (As), and chromium (Cr) were also detected in the soil. Although their concentrations were generally lower than those of copper and zinc, their presence nonetheless indicated a broader pattern of soil contamination associated with mining activities. For example, lead concentrations in some areas exceeded the permissible limits for agricultural soils by a factor of five.

The study revealed significant concentrations of heavy metals, primarily zinc and copper, in grass samples collected from areas adjacent to mining operations. These findings strongly suggest a direct correlation between historical mining activities and the contamination of grazing lands. Spatial analysis indicated higher metal concentrations closer to pollution sources, highlighting the long-term environmental impacts of mining. Notably, copper levels in grass from Firiza reached as high as 9.28 mg/kg, significantly exceeding reference values. Such elevated metal burdens pose substantial risks to human health through the consumption of contaminated animal products. The bioaccumulation of heavy metals in the food chain can lead to chronic health issues in humans, including organ damage, neurological disorders, and increased cancer risk. Given the potential for human exposure, further research is warranted to develop effective remediation strategies and to implement robust food safety measures.

Analysis of sheep milk and cheese samples revealed varying levels of heavy metals. Zinc was the predominant metal detected in both milk and cheese, followed by copper, lead, and cadmium. Notably, cheesemaking significantly influenced mineral distribution, with zinc concentrating in the curd portion. Spatial analysis revealed distinct patterns in heavy metal concentrations, linking elevated levels in certain areas to potential environmental contamination from historical mining activities. For instance, high copper and zinc concentrations were observed in samples from the Ferneziu I area, suggesting a geological influence. Conversely, lead levels were elevated in samples from Firiza, potentially linked to the nearby Aurul settling pond. While the concentrations of nickel, cobalt, arsenic, chromium, and mercury were below detection limits, the presence of other heavy metals, particularly at concentrations approaching or exceeding regulatory limits, raises concerns about food safety and human health. The bioaccumulation of heavy metals in humans through the consumption of contaminated dairy products can lead to a range of health issues, including organ damage, neurological disorders, and increased cancer risk [69]. Further research is needed to fully understand the pathways of heavy metal contamination in the food chain and to develop effective strategies to mitigate these risks [70].

Serum analysis revealed a complex interplay of heavy metal concentrations in sheep. Zinc was the predominant metal, followed by copper and lead, with negligible levels of other elements. Spatial variations in metal concentrations were observed, potentially linked to environmental factors. While overall levels were relatively low, the presence of lead, even at trace amounts, warrants further investigation due to its known toxicity. These findings underscore the importance of continuous monitoring of animal health and the environmental factors influencing metal exposure.

The spatial distribution of heavy metals within the study area exhibited a clear pattern, with concentrations significantly elevated in proximity to the pollution source. This finding underscores the impact of anthropogenic activities on environmental contamination. A strong correlation was observed between heavy metal concentrations and distance from the former Herja mine, suggesting a direct link between mining activities and soil pollution. Furthermore, localized hotspots of contamination were identified, often associated with specific land use practices or geological features. The spatial variability of heavy metals highlights the importance of considering geographical factors when assessing environmental contamination. Understanding these patterns is crucial for identifying areas at higher risk and for developing targeted remediation strategies.

Bioaccumulation factors (BAFs) for heavy metals were calculated across the sheep-based food chain, from soil to dairy products. The results indicate significant variations in BAFs among metals and matrices. Nickel displayed the highest BCF in grass, suggesting a strong plant uptake, while cadmium accumulated efficiently in dairy products. Zinc and copper BCFs were influenced by dietary intake, metabolism, and interactions with other metals. These findings highlight the complex nature of heavy metal bioaccumulation and the potential for human exposure through the consumption of contaminated food products.

## 4. Conclusions

This study revealed significant heavy metal contamination in the Baia Mare area, Romania, particularly copper (Cu) and zinc (Zn), which exceeded permissible limits, especially near the abandoned Herja mine (Cu: up to 2528.20 mg/kg; Zn: up to 1821.40 mg/kg). These findings indicate a direct correlation between historical mining activities and the elevated levels of these metals. Additionally, elevated concentrations of lead (Pb) and cadmium (Cd) were identified in the industrial area of Ferneziu, pointing to industrial emissions as a contributing factor. Analysis of green grass samples confirmed substantial spatial variability, with Zn being the most abundant metal, followed by Cu. Notably, the pronounced patterns of Cu concentration near mining operations raise concerns about the environmental risks associated with heavy metal exposure. While nickel (Ni) and cobalt (Co) showed lower average levels, localized peaks near contamination sources necessitate further investigation into their mobility. Sheep milk and cheese analyses demonstrated consistent concentrations of heavy metals, with Zn showing significant variation, indicating a potential bioaccumulation trend. The findings emphasize the need for monitoring and regulatory compliance concerning these metals in food products. Overall, this research underscores the importance of understanding heavy metal dynamics in the food chain and the implications for human health. Future studies should prioritize soil composition analysis, dietary management in sheep husbandry, and the potential impact of environmental contaminants to develop effective risk assessment and mitigation strategies.

## Figures and Tables

**Figure 1 toxics-12-00752-f001:**
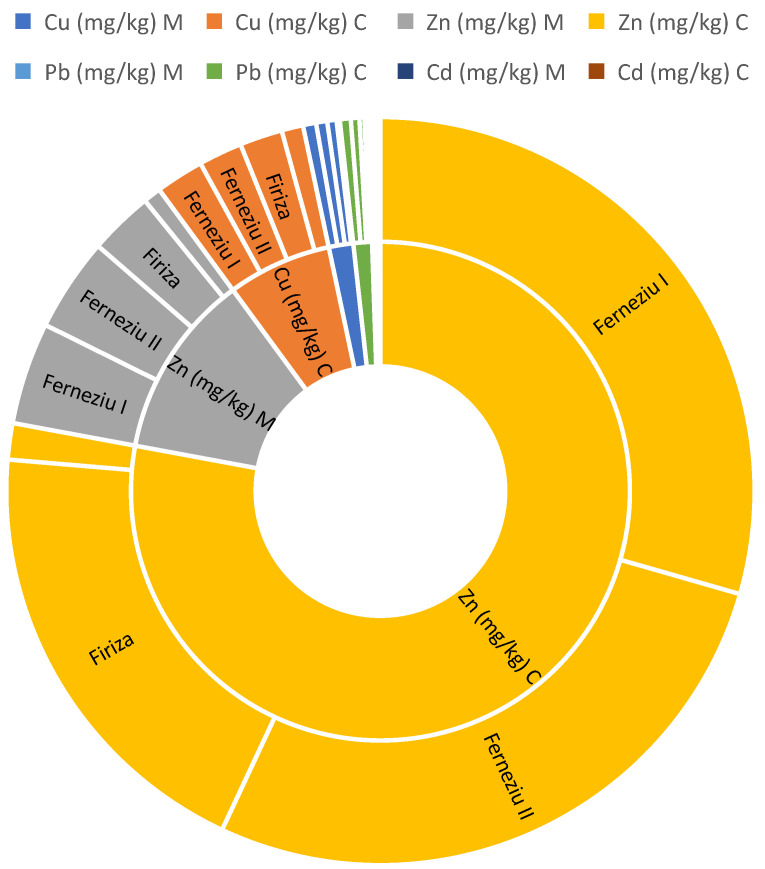
The spatial distribution of detectable elements in sheep milk and cheese samples; comparing mean concentration across different collection areas.

**Figure 2 toxics-12-00752-f002:**
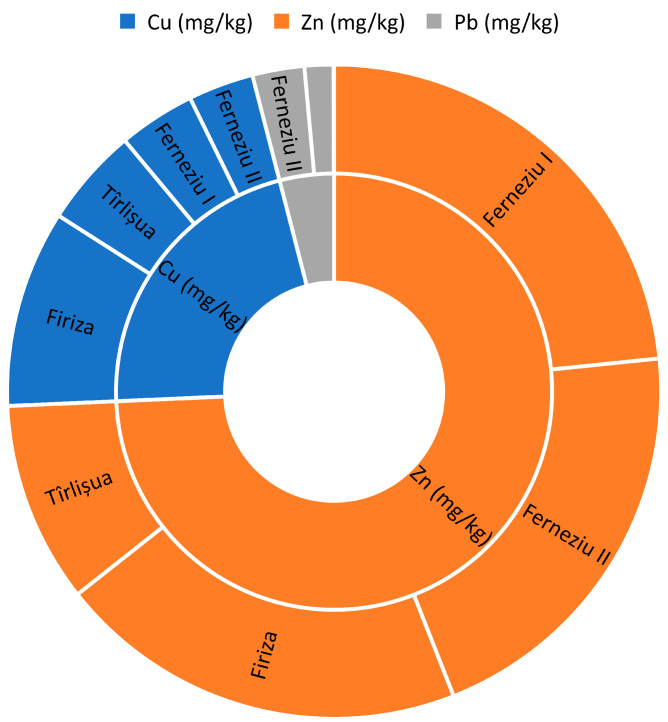
The spatial distribution of detectable elements in sheep serum samples; comparing mean concentration across different collection areas.

**Table 3 toxics-12-00752-t003:** The content of heavy metals in sheep’s milks and cheese from Ferneziu, Firiza (Maramureș area) and Tîrlișua (Bistrița-Năsăud), Romania. (mg/kg WW) (Mean ± standard deviation) (n = 36).

**Areas** **Sample Code** **Year of Harvest**	**Distance from the Source of Pollution (** **~) km**	**Sampling** **Depth** **(Surface)**	**Cu** M.P.L.	**Zn** M.P.L.	**Pb** M.P.L.	**Cd** M.P.L.	**Ni** M.P.L.	**Co** M.P.L.	**As** M.P.L.	**Cr** M.P.L.	**Hg** M.P.L.
			Regulatory guidelines for maximum permissible levels of arsenic and heavy metals in food products (milk) [58]								
Alert threshold		–	0.5 mg/kg	5 mg/kg	0.1 mg/kg	0.01 mg/kg	–	–	0.1 mg/kg	–	0.01
		–									
Intervention threshold		–	Regulatory guidelines for maximum permissible levels of arsenic and heavy metals in food products (cheese) [58]								
		–	2.5 mg/kg	25 mg/kg	0.5 mg/kg	0.05 mg/kg	–	–	0.15 mg/kg	–	–
Sheep’s milks and cheese samples exposed to anthropogenic sources of heavy metals pollution											
Ferneziu	Near (~) 10/12 km to the Herja Mine from Ferneziu										
M_1-2024_ 2024	The sheepfold was located approximately (~) 8.0 km to the former Herja mine in Ferneziu										
			0.16–1.93 0.70 ± 0.63 ^a^	5.04–7.16 6.33 ± 0.75 ^a^	BLD—0.16 0.08 ± 0.06 ^d^	BLD—0.08 0.03 ± 0.04 ^a^	BLD	BLD	BLD	BLD	BLD
C_1-2024_ 2024			1.65–4.02 2.92 ± 0.83 ^α^	25.74–59.87 38.36 ± 14.65 ^α^	BLD—0.19 0.19 ± 0.11 ^ζ^	BLD—0.12 0.04 ± 0.04 ^δ^	BLD	BLD	BLD	BLD	BLD
M_2-2024_ 2024	The sheepfold was located approximately (~) 11.5 km from the former Herja mine in Ferneziu										
			0.32–1.02 0.55 ± 0.26 ^c^	2.63–6.72 4.18 ± 1.55 ^c^	BLD—0.10 0.04 ± 0.04 ^f^	BLD—0.03 0.02 ± 0.01 ^b^	BLD	BLD	BLD	BLD	BLD
C_2-2024_ 2024			0.31–2.81 1.91 ± 1.19 ^ζ^	18.37–42.20 29.94 ± 7.73 ^δ^	BLD—0.65 0.31 ± 0.22 ^ε^	BLD—0.09 0.05 ± 0.03 ^β^	BLD	BLD	BLD	BLD	BLD
Ferneziu	Near (~6/7 km) the Herja Mine in Ferneziu										
M_3-2024_ 2024	The sheepfold was located approximately (~) 5.5 km from the former Herja mine in Ferneziu										
			0.27–0.55 0.44 ± 0.09 ^e^	1.98–10.56 5.35 ± 2.92 ^b^	BLD—0.25 0.13 ± 0.08 ^b^	BLD—0.05 0.02 ± 0.02 ^b^	BLD	BLD	BLD	BLD	BLD
C_3-2024_ 2024			1.37–2.55 2.17 ± 0.42 ^δ^	8.77–58.49 33.29 ± 20.41 ^β^	BLD—0.65 0.49 ± 0.25 ^β^	BLD—0.05 0.05 ± 0.03 ^β^	BLD	BLD	BLD	BLD	BLD
M_4-2024_ 2024	The sheepfold was located approximately (~) 7.5 km from the former Herja mine in Ferneziu										
			0.29–1.49 0.67 ± 0.43 ^b^	2.59–5.56 4.06 ± 1.19 ^d^	BLD—0.25 0.10 ± 0.10 ^d^	BLD—0.05 0.03 ± 0.02 ^a^	BLD	BLD	BLD	BLD	BLD
C_4-2024_ 2024			1.05–2.72 2.21 ± 0.62 ^γ^	19.09–50.64 30.55 ± 12.11 ^γ^	BLD—0.58 0.37 ± 0.21 ^δ^	BLD—0.10 0.06 ± 0.02 ^α^	BLD	BLD	BLD	BLD	BLD
Firiza	Near (~17 km) the Aurul settling pond mining (decant pond) in Tăuții de Sus										
M_5-2024_ 2024	The sheepfold was located approximately (~) 16.5 km from the Aurul settling pond mining (decant pond) in Tăuții de Sus										
			0.28–0.79 0.44 ± 0.17 ^b^	2.18–6.13 3.79 ± 1.11 ^e^	BLD—0.21 0.14 ± 0.07 ^a^	BLD—0.04 0.02 ± 0.01 ^b^	BLD	BLD	BLD	BLD	BLD
C_5-2024_ 2024			0.79–3.19 2.01 ± 0.74 ^ε^	18.36–39.24 27.35 ± 7.76 ^ε^	0.46–0.76 0.58 ± 0.10 ^α^	BLD–0.10 0.03 ± 0.03 ^δ^	BLD	BLD	BLD	BLD	BLD
M_6-2024_ 2024	The sheepfold was located approximately (~) 17.0 km from the Aurul settling pond mining (decant pond) in Tăuții de Sus										
			0.32–0.92 0.51 ± 0.28 ^d^	1.49–2.58 2.08 ± 0.51 ^f^	BLD–0.24 0.12 ± 0.10 ^c^	0.01–0.02 0.02 ± 0.01 ^b^	BLD	BLD	BLD	BLD	BLD
C_6-2024_ 2024			2.15–2.81 2.42 ± 0.28 ^β^	3.27–16.46 12.77 ± 6.37 ^ζ^	0.25–0.63 0.47 ± 0.18 ^γ^	0.01–0.06 0.04 ± 0.02 ^γ^	BLD	BLD	BLD	BLD	BLD
Background sheep’s milks and cheese samples											
Tîrlișua	–										
	–										
M_7-2024_ 2024			0.15–0.16 0.16 ± 0.01 ^f^	0.23–1.56 0.89 ± 0.67 ^g^	0.01–0.06 0.03 ± 0.03 ^f^	BLD	BLD	BLD	BLD	BLD	BLD
C_7-2024_ 2024			0.98–1.15 1.08 ± 0.09 ^η^	1.11–3.23 1.82 ± 1.22 ^η^	0.11–0.21 0.17 ± 0.05 ^η^	BLD	BLD	BLD	BLD	BLD	BLD
Sig.			***	***	***	***	–	–	–	–	–
Sheep’s milks and cheese samples exposed to anthropogenic sources of heavy metals pollution											
Anastasio et al., (2006) (Sheep milk µg/g) [59]			–	–	0. 11–0.28	0.05–0.10	–	–	–	0.06–0.40	0.002–0.005
Anastasio et al., (2006) (Fresh chesse µg/g) [59]			–	–	0.13–1.15	0.05–0.35	–	–	–	0.40–0.53	0.006–0.0017
Al Sidawi et al., (2021) (mg/L) [5]			0.120–0.592	2.223–4.294	–	<0.001	0.001–0.017	0.002–0.007	–	0.001–0.004	0.004–0.013
Background sheep’s milks and cheese samples											
Magdas et al., (2019) (Chesse mg/kg) [60]			–	<0.001–62.84	–	–	–	–	0.006–0.753	–	–
Martínez-Morcillo et al., (2024) (Sheep milk µg/kg) [2]			47.2–476	1930–8580	19.7–32.7	–	–	–	–	–	–
Suhaj et al., (2008) (chesse mg/kg) [54]			0.686–1.53	–	–	–	0.203–0.501	–	–	0.111–0.373	0.0001–0.0006

Average value ± standard deviation (n = 36). WW = wet weight. Roman letters are the significance of the difference (*p* ≤ 0.005) regardless of the area of sample collection for the milk samples. Greek letters represent the significance of the difference (*p* ≤ 0.005) regardless of the sample collection area, for the cheese samples. Samples were evaluated against the maximum permissible limits (MPLs) established by regulatory threshold levels for arsenic and heavy metals in foods products (mg/kg) [58]; BLD = Below the detection limit (LoQ): LoQ for Pb: 0.231 µg/L, LoQ for Cd: 0.069 µg/L, LoQ for Co: 0.136 µg/L, LoQ for As: 0.743 µg/L; LoQ for Hg 0.1379 µg/L. *** = shows a significant difference between the analyzed variants. significance of the difference (*p* ≤ 0.005).

**Table 5 toxics-12-00752-t005:** An assessment of bioaccumulation factors for heavy metals in green grass, sheep milk, cheese, and serum.

Sample	Area Sample	Cu	Zn	Pb	Cd	Ni	Co	As	Cr	Hg	TOTAL
Green Grass	Ferneziu I	0.0008	0.0970	0.0094	0.1022	1.1276	–	–	–	–	**1.3370**
	Ferneziu II	0.0006	0.0601	0.0058	0.0882	1.3407	–	–	–	–	**1.4954**
	Firiza	0.0025	0.0685	0.0113	0.2418	4.6530	–	–	–	–	**4.9771**
**TOTAL**		**0.0040**	**0.2255**	**0.0266**	**0.4322**	**7.1214**	**–**	**–**	**–**	**–**	**7.8097**
Milk	Ferneziu I	0.0023	0.0072	0.0004	0.1022	–	–	–	–	–	**0.1121**
	Ferneziu II	0.0017	0.0045	0.0005	0.0882	–	–	–	–	–	**0.0949**
	Firiza	0.0019	0.0037	0.0025	0.2418	–	–	–	–	–	**0.2499**
**TOTAL**		**0.0059**	**0.0154**	**0.0034**	**0.4322**	**–**	**–**	**–**	**–**	**–**	**0.4969**
Cheese	Ferneziu I	0.0084	0.0482	0.0020	0.0240	–	–	–	–	–	**0.0826**
	Ferneziu II	0.0068	0.0307	0.0018	0.0324	–	–	–	–	–	**0.0324**
	Firiza	0.0087	0.0256	0.0101	0.0613	–	–	–	–	–	**0.1057**
**TOTAL**		**0.0239**	**0.1045**	**0.0139**	**0.1177**	**–**	**–**	**–**	**–**	**–**	**0.2207**
Serum	Ferneziu I	0.0008	0.0021	0.0007	–	–	–	–	–	–	**0.0036**
	Ferneziu II	0.0006	0.0013	0.0007	–	–	–	–	–	–	**0.0026**
	Firiza	0.0025	0.0015	–	–	–	–	–	–	–	**0.0040**
**TOTAL**		**0.0039**	**0.049**	**0.0014**	**–**	**–**	**–**	**–**	**–**	**–**	**0.0102**
**Sample**	**Area Sample**	**Heavy Metal Concentration**									
Green Grass	Ferneziu I	Ni > Cd > Zn > Pb > Cu									
	Ferneziu II	Ni > Cd > Zn > Pb > Cu									
	Firiza	Ni > Cd > Zn > Pb > Cu									
		**Ni > Cd > Zn > Pb > Cu**									
Milk	Ferneziu I	Cd > Zn > Cu > Pb									
	Ferneziu II	Cd > Zn > Cu > Pb									
	Firiza	Cd > Zn > Pb > Cu									
		**Cd > Zn > Cu > Pb**									
Cheese	Ferneziu I	Zn > Cd > Cu > Pb									
	Ferneziu II	Cd > Zn > Cu > Pb									
	Firiza	Cd > Zn > Pb > Cu									
		**Cd > Zn > Cu > Pb**									
Serum	Ferneziu I	Zn > Cu > Pb									
	Ferneziu II	Zn > Pb > Cu									
	Firiza	Cu > Zn									
		**Zn > Cu > Pb**									

## Data Availability

The manuscript contains a detailed presentation of the study’s novel findings. Appendix A offer further supporting data. For more information, please contact the corresponding author.

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
