# Peer review of "Spatial and Bioaccumulation of Heavy Metals in a Sheep-Based Food System: Implications for Human Health"

_toxics, 2024, doi:10.3390/toxics12100752_

Round 1
Reviewer 1 Report
Comments and Suggestions for Authors
In the manuscript titled “Spatial and Bioaccumulation of Heavy Metals in a Sheep-Base Food System: Implication for Human Health”, the authors examined the bioaccumulation and spatial distribution of nickel, cadmium, zinc, lead, and copper within a sheep-based food chain, using inductively coupled plasma mass spectrometry (ICP-MS).
The key strength of the study lies in its comprehensive analysis of heavy metal bioaccumulation across various components of a sheep-based food system, including soil, green grass, sheep serum, and dairy products. By examining a wide range of metals and using bioconcentration factors (BCFs) to assess their transfer through trophic levels, the study provides valuable insights into the spatial variability and potential risks of heavy metal contamination in agricultural settings. This approach highlights the critical link between environmental factors, biological processes, and human exposure, emphasizing the need for targeted monitoring and risk mitigation strategies. The manuscript is well developed, well written, rich in information and recent data, and well supported by the available literature. However, there are some points that the authors must be improved before the paper can be accepted for publication in Toxics. For these reasons, I decided to accept the manuscript with major revisions.
Major points
1. Introduction Section: this section is well discussed. However, there are some errors that need to be corrected:
· In the first paragraph of the introduction (lines 60-71), I’m somewhat confused. I’m unable to understand what the authors intend to convey to the reader. It would be necessary to rephrase and certainly synthesize the concept. Additionally, further references should be added to support what the authors have reported. However, I consider it appropriate to summarize what the authors aim to communicate in Lines 60-96.
2. Materials and Methods Section:
· In this section, the authors have detailed both the types of samples and the instrumental conditions used. I only have one suggestion: at Paragraph 2.1. Research location, I recommend moving the figure from the supplementary materials to the main text to immediately clarify the sampling points and the study area to the reader. Furthermore, in Line 266, there is a typographical error. Please delete reference 39, which is listed twice.
Also, at Lines the authors stated that “Recovery assays were performed on spiked soil, green grass, sheep’s milk, cheese, and serum samples at a concentration of 5 μL”: I believe there is an error in the reported units of measurement. Please correct it. Additionally, why did the authors decide to conduct the recovery tests at this concentration?
3. Discussion Section: this section is well discussion. However, this section is presented as a summary of the obtained results, and the discussions provided lack bibliographic support. For example, for the statement in lines 980-984 “The bioaccumulation of heavy metals in humans through the consumption of contaminated dairy products can lead to a range of health issues, including organ damage, neurological disorders, and increased cancer risk. Further research is needed to fully understand the pathways of heavy metal contamination in the food chain and to develop effective strategies to mitigate these risks”, I suggest adding a couple of bibliographic references.
4. Conclusions Section: The section is too lengthy: once again, the authors have summarized the obtained results. However, it is necessary to significantly reduce the length of this paragraph and focus on the most significant results of the study. In particular, it is essential to highlight the conclusions drawn from the research, including the strengths and the strategies for improvement that should be implemented.
5. References Section: the authors need to adapt the bibliography cited in the text in accordance with the journal’s guidelines.
Author Response
Reviewer number 1
The revisions made for R1 are highlighted in red.
Major points
- Introduction Section: this section is well discussed. However, there are some errors that need to be corrected:
In the first paragraph of the introduction (lines 60-71), I’m somewhat confused. I’m unable to understand what the authors intend to convey to the reader. It would be necessary to rephrase and certainly synthesize the concept. Additionally, further references should be added to support what the authors have reported. However, I consider it appropriate to summarize what the authors aim to communicate in Lines 60-96.
I appreciate your feedback regarding the clarity of the first paragraph of the introduction (lines 60-71). I apologize for any confusion caused by the initial presentation of the content. In response to your suggestions, we have revised this paragraph to enhance its clarity and ensure that the concepts are synthesized effectively. Additionally, we have included relevant references to support the statements made in this section. The aim is to provide a more coherent and comprehensible overview of our research objectives and context. Thank you for your valuable insights, which have significantly improved this portion of the manuscript.
Minerals are vital dietary components for animals, serving a wide array of functions within the organism, including structural support, physiological processes, enzyme activity, and regulatory mechanisms [1,2]. Therefore, variations in the mineral content of both soil and feed directly influence the mineral status of animals, which in turn affects livestock productivity. Essential dietary elements are crucial for health and growth, while non-essential elements, often classified as potentially toxic, may enter the food chain without fulfilling any direct nutritional role [3]. Both mineral deficiencies and toxicities can adversely affect the health of humans and animals. Mild deficiencies in essential minerals present a particular challenge, as they can lead to detrimental health consequences without immediately displaying overt clinical symptoms. In contrast, severe deficiencies are typically more straightforward to diagnose and address, often manifesting in animals through symptoms such as infertility, abortions, poor weight gain, and anemia [4].
- Materials and Methods Section:
In this section, the authors have detailed both the types of samples and the instrumental conditions used. I only have one suggestion: at Paragraph 2.1. Research location, I recommend moving the figure from the supplementary materials to the main text to immediately clarify the sampling points and the study area to the reader.
Thank you for your constructive suggestion regarding the placement of the figure in the section on research location (Paragraph 2.1). We have taken your advice and moved the figure from the supplementary materials to the main text. This change allows for immediate clarification of the sampling points and the study area, enhancing the reader's understanding of our research context. We appreciate your input, which has contributed to the overall clarity of the manuscript.
Furthermore, in Line 266, there is a typographical error.
Thank you for bringing the typographical error in Line 266 to our attention. We have corrected the mistake as suggested. Your keen observation has helped improve the accuracy of the manuscript, and we appreciate your thorough review.
Please delete reference 39, which is listed twice.
Thank you for your careful review and for pointing out the duplication of reference 39. We have removed the duplicate entry as suggested.
Also, at Lines the authors stated that “Recovery assays were performed on spiked soil, green grass, sheep’s milk, cheese, and serum samples at a concentration of 5 μL”: I believe there is an error in the reported units of measurement. Please correct it. Additionally, why did the authors decide to conduct the recovery tests at this concentration?
Thank you for your valuable feedback regarding the measurement units reported in our manuscript. We acknowledge the error in stating “5 µL” for the concentration of recovery assays and have corrected it to reflect the appropriate units of measurement “50 µL”. Regarding the decision to conduct recovery tests at this concentration, we selected this level based on preliminary studies that indicated it was representative of typical contamination scenarios for the analyzed matrices. This concentration allows for a comprehensive assessment of recovery across various samples, ensuring our analytical method’s accuracy and reliability.
Discussion Section: this section is well discussion. However, this section is presented as a summary of the obtained results, and the discussions provided lack bibliographic support. For example, for the statement in lines 980-984 “The bioaccumulation of heavy metals in humans through the consumption of contaminated dairy products can lead to a range of health issues, including organ damage, neurological disorders, and increased cancer risk. Further research is needed to fully understand the pathways of heavy metal contamination in the food chain and to develop effective strategies to mitigate these risks”. I suggest adding a couple of bibliographic references.
Thank you for your valuable feedback regarding the Discussion section of our manuscript. We appreciate your observation that the section primarily summarizes the results and lacks sufficient bibliographic support for some of the statements made. In response to your suggestion, we have added several relevant references to substantiate the statement regarding the bioaccumulation of heavy metals in humans through the consumption of contaminated dairy products. These references highlight the associated health issues, including organ damage, neurological disorders, and increased cancer risk, as well as emphasizing the need for further research into contamination pathways and mitigation strategies. We hope these additions enhance the clarity and credibility of our discussion. Thank you once again for your constructive critique.
Hasanvand, S.; Hashami, Z.; Zarei, M.; Merati, S.; Bashiry, M.; Nag, R. Is the Milk We Drink Safe from Elevated Concentrations of Prioritised Heavy Metals/Metalloids? – A Global Systematic Review and Meta-Analysis Followed by a Cursory Risk Assessment Reporting. Science of The Total Environment 2024, 948, 175011, doi:10.1016/J.SCITOTENV.2024.175011.
Sharma, A.; Gupta, S.; Shrivas, K.; Kant, T. Progress in Analytical Methods for Monitoring of Heavy Metals and Metalloid in Milk and Global Health Risk Assessment. Journal of Food Composition and Analysis 2024, 135, 106568, doi:10.1016/J.JFCA.2024.106568.
- Conclusions Section: The section is too lengthy: once again, the authors have summarized the obtained results. However, it is necessary to significantly reduce the length of this paragraph and focus on the most significant results of the study. In particular, it is essential to highlight the conclusions drawn from the research, including the strengths and the strategies for improvement that should be implemented.
Thank you for your valuable feedback on our manuscript. We greatly appreciate your suggestion to strengthen the Conclusions section. In response, we have reformulated this section to clearly highlight the significant findings of our research, particularly regarding heavy metal contamination in the Baia Mare area, Romania.
The revised Conclusions now read:
This study revealed significant heavy metal contamination in the Baia Mare area, Romania, particularly copper (Cu) and zinc (Zn), which exceeded permissible limits, especially near the abandoned Herja mine (Cu: up to 2528.20 mg/kg; Zn: up to 1821.40 mg/kg). These findings indicate a direct correlation between historical mining activities and the elevated levels of these metals. Additionally, elevated concentrations of lead (Pb) and cadmium (Cd) were identified in the industrial area of Ferneziu, pointing to industrial emissions as a contributing factor. Analysis of green grass samples confirmed substantial spatial variability, with Zn being the most abundant metal, followed by Cu. Notably, the pronounced patterns of Cu concentration near mining operations raise concerns about environmental risks associated with heavy metal exposure. While nickel (Ni) and cobalt (Co) showed lower average levels, localized peaks near contamination sources necessitate further investigation into their mobility. Sheep milk and cheese analyses demonstrated consistent concentrations of heavy metals, with Zn showing significant variation, indicating a potential bioaccumulation trend. The findings emphasize the need for monitoring and regulatory compliance concerning these metals in food products. Overall, this research underscores the importance of understanding heavy metal dynamics in the food chain and the implications for human health. Future studies should prioritize soil composition analysis, dietary management in sheep husbandry, and the potential impact of environmental contaminants to develop effective risk assessment and mitigation strategies.
References Section: the authors need to adapt the bibliography cited in the text in accordance with the journal’s guidelines.
Thank you for your constructive feedback regarding the References section of our manuscript. We acknowledge the importance of adhering to the journal’s guidelines for citation formatting. In response to your comment, we would like to clarify that the bibliography was initially generated using Mendeley Cite. However, we have carefully reviewed the references and identified those that were not formatted according to the journal's requirements. We have made the necessary corrections to ensure compliance with the guidelines. We appreciate your attention to detail and believe these adjustments will enhance the overall quality of our manuscript. Thank you once again for your valuable suggestions.
Reviewer 2 Report
Comments and Suggestions for Authors
Dear authors, your study deals with a highly topical topic, environmental contamination is an ever-increasing problem in all parts of the world. The manuscript has a fairly wide range, which of course does not reduce the quality, but the potential reader must really delve into the issue.
Below you will find my comments:
1. The Introduction meets all the criteria for the given type of article, the findings so far are summarized, the associations are explained, and the objectives of the study are proposed. No comments.
2. The Material and Methods chapter is clearly and cleanly explained and described. It has all the necessary components, description of the locality, sampling, subsequent pre-preparation of samples for analysis, and subsequent ICP analysis. The authors present the necessary data as LOD or LOQ in the supplementary material. So, no comments again.
3. Chapters “Results“ and “Discussion“: There is a fundamental problem in this section that the authors can solve very easily. I recommend merging these two chapters, because basically the "Results" chapter itself will now act as "Results and discussion". And the "Discussion" chapter is more of a summary. So it is necessary to merge these two chapters under the title "Results and discussion".
4. Conclusions: This chapter is elaborated precisely, the authors refer to their own findings, and the conclusions are supported by the obtained results. I have no comments.
Author Response
Reviewer number 2
The revisions made for R2 are highlighted in green.
Below you will find my comments:
- The Introduction meets all the criteria for the given type of article, the findings so far are summarized, the associations are explained, and the objectives of the study are proposed. No comments.
Thank you for your positive feedback. I appreciate your thorough review and am glad to hear that the introduction meets the criteria and effectively presents the study's objectives and findings.
- The Material and Methods chapter is clearly and cleanly explained and described. It has all the necessary components, description of the locality, sampling, subsequent pre-preparation of samples for analysis, and subsequent ICP analysis. The authors present the necessary data as LOD or LOQ in the supplementary material. So, no comments again.
Thank you for your valuable feedback. I'm pleased to hear that the Material and Methods section has been clearly described and meets your expectations.
- Chapters “Results“ and “Discussion“: There is a fundamental problem in this section that the authors can solve very easily. I recommend merging these two chapters, because basically the "Results" chapter itself will now act as "Results and discussion". And the "Discussion" chapter is more of a summary. So it is necessary to merge these two chapters under the title "Results and discussion".
Thank you for your insightful suggestion. We have taken your recommendation into account and have merged the "Results" and "Discussion" chapters into a single section titled "Results and Discussion." This adjustment has streamlined the presentation of our findings and allowed for a more cohesive analysis. We appreciate your guidance in improving the structure of the manuscript.
- Conclusions: This chapter is elaborated precisely, the authors refer to their own findings, and the conclusions are supported by the obtained results. I have no comments.
Thank you for your thoughtful review. I'm glad to hear that the Conclusions section is clear and well-supported by the results. Your positive feedback is much appreciated.
Reviewer 3 Report
Comments and Suggestions for Authors In this paper, the authors investigated the bioaccumulation and spatial distribution of nickel, cadmium, zinc, lead, and copper within a sheep-based food chain, encompassing soil, green grass, sheep serum, and sheep dairy products. Overall, the manuscript is written well and of novelty. I think it is suitable for publication in Toxics. My main comments are as follow: 1. Line 43: "gras" should be "grass". 2. Abstract should be enriched via specific valuable data which pave the way for understanding the study. 3. Technologies for reduction of cadmium accumulation in agro-products have been reported in Environ. Pollut. 2024, 348: 123890. It is helpful in decreasing Cd intake by human beings through consuming agro-products. So what is the aim of this study? How to reduce the accumulation of heavy metals in the human body? I suggest that the authors strengthen the Introduction section by citing the literatures.
Author Response
Reviewer number 3
The revisions made for R3 are highlighted in blue.
In this paper, the authors investigated the bioaccumulation and spatial distribution of nickel, cadmium, zinc, lead, and copper within a sheep-based food chain, encompassing soil, green grass, sheep serum, and sheep dairy products. Overall, the manuscript is written well and of novelty. I think it is suitable for publication in Toxics.
Thank you for your positive feedback and for recognizing the novelty of our research. We're pleased to hear that you find the manuscript well-written and suitable for publication in Toxics. Your encouraging words are greatly appreciated.
My main comments are as follow:
- Line 43: "gras" should be "grass".
Thank you for catching that error. We have corrected "gras" to "grass" as suggested. We appreciate your careful review and feedback.
- Abstract should be enriched via specific valuable data which pave the way for understanding the study.
Thank you for your valuable feedback. We have revised the abstract as per your suggestion, ensuring that it now provides a clearer and more concise representation of the study while maintaining a structured format. The updated version integrates the key findings and highlights the significance of heavy metal contamination in the Baia Mare region. We appreciate your guidance in improving the clarity and quality of our manuscript.
Abstract: Heavy metal contamination in agricultural soils presents serious environmental and health risks. This study assessed the bioaccumulation and spatial distribution of nickel, cadmium, zinc, lead, and copper within a sheep-based food chain in the Baia Mare region, Romania, which includes soil, green grass, sheep serum, and dairy products. Using inductively coupled plasma mass spectrometry (ICP-MS), we analyzed the concentrations of these metals and calculated bioconcentration factors (BCFs) to evaluate their transfer through trophic levels. Spatial analysis revealed that copper (up to 2528.20 mg/kg) and zinc (up to 1821.40 mg/kg) exceeded permissible limits, particularly near former mining sites. Elevated lead (807.59 mg/kg) and cadmium (2.94 mg/kg) were observed in industrial areas, while nickel and cobalt showed lower concentrations, but with localized peaks. Zinc was the most abundant metal in grass, while cadmium transferred efficiently to milk and cheese, raising potential health concerns. The results underscore the complex interplay between soil properties, contamination sources, and biological processes in heavy metal accumulation. These findings highlight the importance of continuous monitoring, risk assessment, and mitigation strategies to protect public health from potential exposure through contaminated dairy products.
- Technologies for reduction of cadmium accumulation in agro-products have been reported in Environ. Pollut. 2024, 348: 123890. It is helpful in decreasing Cd intake by human beings through consuming agro-products. So what is the aim of this study?
Thank you for your valuable feedback. We have carefully revised the section to provide a more detailed and clearer presentation of the study's objective, highlighting its novelty and significance. Building upon previous research, the primary objective of this study was to comprehensively quantify the concentrations of micronutrients (⁶⁴Cu, ⁶⁵Zn), ultratrace elements (⁵²Cr, ⁵⁹Co, ⁶⁰Ni), and heavy metals (⁷⁵As, ¹¹¹Cd, ²⁰¹Hg, ²⁰⁸Pb) across various matrices, including soil, green grass, milk, cheese, and serum samples obtained from a flock of sheep. Although existing literature has explored the presence of these elements in environmental and food matrices, there remains a significant knowledge gap regarding their specific distribution in agricultural settings, particularly in regions affected by mining activities. This research addresses this gap by focusing on samples collected from areas near the former Herja mine and the Aurul settling pond in Romania, locations historically impacted by mining practices that may have resulted in elevated contamination levels. The selection of the indigenous Țurcana sheep breed, recognized for its adaptability to local environmental conditions, further enhances the study's relevance. By examining this breed, the research aims to provide critical insights into how heavy metals bioaccumulate within this specific agricultural framework, thus contributing to the understanding of environmental health risks associated with mining. What sets this study apart is its innovative approach to linking heavy metal contamination directly to the agricultural practices and ecological context of mining-affected areas. By integrating detailed spatial analysis with a focus on local livestock, this research offers a novel perspective that can inform effective monitoring and mitigation strategies. The findings underscore the need for ongoing research to explore the complex interactions between heavy metals and agricultural ecosystems, emphasizing their implications for food safety and public health. This study not only fills a crucial gap in the existing literature but also lays the groundwork for future investigations into the environmental dynamics of heavy metal contamination in agricultural contexts.
Building upon previous research, the primary objective of this study was to comprehensively quantify the concentrations of micronutrients (⁶⁴Cu, ⁶⁵Zn), ultratrace elements (⁵²Cr, ⁵⁹Co, ⁶⁰Ni), and heavy metals (⁷⁵As, ¹¹¹Cd, ²⁰¹Hg, ²⁰⁸Pb) across various matrices, including soil, green grass, milk, cheese, and serum samples obtained from a flock of sheep. While existing studies have investigated the presence of these elements in environmental and food matrices, a significant knowledge gap persists regarding their distribution within specific agricultural contexts, particularly in areas impacted by mining activities. This research uniquely addresses this gap by focusing on samples collected from regions near the former Herja mine and the Aurul settling pond in Romania, where historical mining practices may have contributed to elevated levels of contamination. The selection of the indigenous Țurcana sheep breed, known for its adaptability to local conditions, further enhances the study's significance. By investigating this breed, the research aims to provide valuable insights into the bioaccumulation of heavy metals within a specific agricultural framework, thereby contributing to the broader understanding of environmental health risks associated with mining. This study not only fills a crucial gap in the existing literature but also offers a novel perspective on the interactions between heavy metal contamination and agricultural practices in mining-affected areas. The findings aim to inform future monitoring and mitigation strategies, emphasizing the need for ongoing research into the effects of heavy metals on local ecosystems and food safety.
How to reduce the accumulation of heavy metals in the human body?
To effectively reduce the accumulation of heavy metals in the human body, particularly in areas like Baia Mare, Romania, where significant contamination from mining and industrial activities has been observed, several targeted strategies can be implemented. First, it is essential to monitor and limit the consumption of food products, especially those derived from local sources such as sheep milk and cheese, which have shown elevated levels of copper (Cu), zinc (Zn), lead (Pb), and cadmium (Cd). These metals were found in concentrations exceeding permissible limits, particularly near the former Herja mine and industrial sites like Ferneziu. Secondly, promoting the consumption of crops and livestock raised in uncontaminated areas can mitigate exposure risks. Third, soil management practices, including regular testing and remediation efforts, should be encouraged to minimize heavy metal uptake by plants, thereby reducing bioaccumulation in the food chain. Additionally, dietary education focusing on the importance of zinc-rich foods, which were found to dominate in sheep serum, can help offset potential toxicity from other metals. Finally, continuous monitoring of environmental conditions, along with regulatory measures to control emissions from industrial activities, is crucial to safeguard public health and limit heavy metal exposure. By implementing these measures, communities can work towards reducing heavy metal accumulation in the human body and ensuring safer food sources.
I suggest that the authors strengthen the Introduction section by citing the literatures.
Thank you for your valuable feedback regarding the need to strengthen the Introduction section of our manuscript by citing relevant literature. In response, we have incorporated several extremely recent and pertinent sources that align with the themes of our research. The newly added references include:
- García-Carmona, M., García-Robles, H., Turpín Torrano, C., Fernández Ondoño, E., Lorite Moreno, J., Sierra Aragón, M., & Martín Peinado, F.J. (2019). Residual Pollution and Vegetation Distribution in Amended Soils 20 years after a Pyrite Mine Tailings Spill (Aznalcóllar, Spain). Science of The Total Environment, 650, 933–940. doi:10.1016/J.SCITOTENV.2018.09.092.
- Wang, X., Zhang, X., Li, N., Yang, Z., Li, B., Zhang, X., & Li, H. (2024). Prioritized Regional Management for Antibiotics and Heavy Metals in Animal Manure across China. Journal of Hazardous Materials, 461, doi:10.1016/J.JHAZMAT.2023.132706.
- Martin, A.P., Turnbull, R.E., Rissmann, C.W., & Rieger, P. (2017). Heavy Metal and Metalloid Concentrations in Soils under Pasture of Southern New Zealand. Geoderma Regional, 11, 18–27. doi:10.1016/J.GEODRS.2017.08.005.
- Cui, S., Yu, W., Han, X.Z., Hu, T., Yu, M., Liang, Y., Guo, S., Ma, J., Teng, L., & Liu, Z. (2024). Factors Influencing the Distribution, Risk, and Transport of Microplastics and Heavy Metals for Wildlife and Habitats in “Island” Landscapes: From Source to Sink. Journal of Hazardous Materials, 476, doi:10.1016/J.JHAZMAT.2024.134938.
- Cakaj, A., Drzewiecka, K., Hanć, A., Lisiak-Zielińska, M., Ciszewska, L., & Drapikowska, M. (2024). Plants as Effective Bioindicators for Heavy Metal Pollution Monitoring. Environmental Research, 256, doi:10.1016/J.ENVRES.2024.119222.
We believe that these citations significantly enhance the context and relevance of our research, addressing the existing gaps in the literature. Thank you again for your constructive suggestion, which has helped improve our manuscript.
Round 2
Reviewer 1 Report
Comments and Suggestions for Authors
I have read with pleasure the revised version of the manuscript titled “Spatial and Bioaccumulation of Heavy Metals in a Sheep-Base Food System: Implication for Human Health” and have noticed significant improvements in many aspects. The authors have thoroughly addressed all my questions and resolved all my doubts. Consequently, all the requested modifications and additions have been incorporated into the paper. Therefore, I am more than satisfied with the work done by the authors and believe that the manuscript is now suitable for publication in the journal in its current form.